# Exotic symmetries, duality, and fractons
# in $2+1$-dimensional quantum field theory

**Nathan Seiberg and Shu-Heng Shao**

School of Natural Sciences, Institute for Advanced Study, Princeton, NJ 08540, USA

## Abstract

We discuss nonstandard continuum quantum field theories in $2+1$ dimensions. They exhibit exotic global symmetries, a subtle spectrum of charged excitations, and dualities similar to dualities of systems in $1+1$ dimensions. These continuum models represent the low-energy limits of certain known lattice systems. One key aspect of these continuum field theories is the important role played by discontinuous field configurations. In two companion papers, we will present $3+1$-dimensional versions of these systems. In particular, we will discuss continuum quantum field theories of some models of fractons.

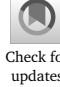
# 1  Introduction

This paper is the first in a sequence of three papers (the other papers are [1, 2]). Here we will study systems in $2+1$ dimensions and in [1, 2] we will consider similar systems in $3+1$ dimensions. (A followup paper [3] explores additional models.) The goal of these papers is to present a continuum quantum field theory perspective of some lattice models studied in the condensed matter literature, and in particular of models of fractons. There is an extensive literature on this subject and it is reviewed nicely in [4, 5]. These reviews includes also many references to the original papers. Below we will refer to the specific papers relevant to each of the topics we discuss.

The main characteristics for a large class of fracton models include:

- The spectrum includes massive particles (fractons) of restricted mobility. Some particles are completely immobile, or can move only along a line, or along a plane. In our treatment of these models we will focus on the low-energy theory. It does not include such dynamical excitations. However, the effect of these massive particles is captures by defects, whose locations are restricted. These defects can be thought of as deformations of the Hamiltonian along lines (or strips) stretched along the time directions.

- The logarithm of the ground state degeneracy of a quantum system is its entropy. Typically it is proportional to the volume of the system, or it is a finite number. The gapped fracton models have a ground state degeneracy proportional to the length of the system.[1] Not only is such behavior surprising from a continuum quantum field theory point of view, it is also infinite in the continuum limit. Specifically, in the X-cube model [7], on a cubic lattice (with periodic boundary conditions) with $L^x$, $L^y$, $L^z$ sites in the three directions, the logarithm of the ground state degeneracy is $2L^x + 2L^y + 2L^z - 3$ and

---

[1]In some cases, as in Haah code [6], the dependence on the size is more complicated.

therefore it diverges in the continuum limit in which $L^x, L^y, L^z \to \infty$. Therefore, any continuum field theory of this system should have an infinite number of ground states.

- These systems exhibit exotic global symmetries. Some of these global symmetries are known as subsystem symmetries.[2] These are symmetries whose charges act only on a subspace of the total space. Unlike the generalized global symmetries of [9], here the value of the charge varies from subspace to subspace [10]. If the global symmetry group is compact such as $U(1)$ or $\mathbb{Z}_N$, this means that the charge operator can be discontinuous as a function of the position.

These three characteristics seem impossible in the context of continuum quantum field theory. For example, one aspect of the degeneracy is that some observables, e.g. some subsystem symmetry generators, can be diagonalized in the space of ground states and their eigenvalues can change discontinuously from one lattice site to another. Equivalently, low-energy observables depend on physics with arbitrarily high momentum. This means that the low-energy physics (IR) is sensitive to some short distance physics (UV).

Despite these challenges, we will attempt to describe these systems using the framework of continuum quantum field theory. Usually, a continuum quantum field theory gives us a universal description of the low-energy physics, which does not depend on most of the details of the microscopic model. One of our motivations is to find such a universal description.

Another motivation, following a broader view, is the pursuit of learning of quantum field theory in its own right and in particular, exploring whether it can be generalized.

Our discussion here (and in [1,2]) will use a number of new elements:

- Not only will these quantum fields theories not be Lorentz invariant, they will also not be rotational invariant. In this paper we will focus on $2 + 1$-dimensional systems and we will not preserve the full $SO(2)$ rotation symmetry, but only its subgroup of 90 degree rotations, $\mathbb{Z}_4$.[3] (We will denote its irreducible, one-dimensional representations $\mathbf{1}_n$ with $n = 0, \pm 1, 2$ labeling the spin.) We will not impose parity or time reversal symmetries, although many of our models are invariant under them. In addition, we have the continuous translation symmetries both in space and time.

- We will continue the investigation of [10], emphasizing the global symmetries of the systems. The discussion of the symmetries is more general than the specific models that we will study. We will also gauge these global symmetries.

- Perhaps the most significant new element is that we will consider discontinuous fields.[4] The underlying spacetime will be continuous, but we will allow discontinuous field configurations. Starting at short distances with a lattice, all the fields are discontinuous there. In standard systems, the fields in the low-energy description are continuous. Here, they will be more continuous than at short distances but some discontinuities will remain.

Rather than giving a general treatment of arbitrary models, our approach will be to find continuum descriptions of specific lattice systems. We start with a lattice with lattice spacing $a$ with $L^i$ sites in the $i$ direction. (In $2 + 1$-dimensional systems the index $i$ takes values $x$ and $y$ and in $3 + 1$-dimensional systems it takes the values $x$, $y$, and $z$.) The continuum limit

---

[2]Subsystem symmetries had figured earlier in various papers, see e.g. [8].

[3]In [1,2] we will study 3+1-dimensional systems and will preserve only their cubic symmetry group $S_4$ (ignoring parity and time reversal).

[4]It is well known that the dominant configurations contributing to the Euclidean path integral are discontinuous. The suppression due to their infinite action is compensated by the large number of such configurations. We do not see a relation between this fact and the discontinuous configurations that we will study.

is $a \to 0$ (with an appropriate limit of the coupling constants), $L^i \to \infty$, while keeping the physical size of the system $\ell^i = aL^i$ fixed. Then, the low-energy physics is defined to be the physics of modes with finite energy in that limit.

In our examples below, we will find some special states, whose energy scales in the continuum limit like $\frac{1}{a}$. Normally, such states are neglected in the continuum limit. However, we will encounter situations where it is meaningful to study them in the continuum limit. This happens because these states are the lowest energy states that are charged under some global symmetry.

A related discussion applies to the space of fields in our field theories. As we said, we will allow discontinuous field configurations with certain discontinuities. These fields, despite being discontinuous, have finite action. Therefore, such configurations must be included in the functional integral.

We will also find it interesting to study discontinuous field configurations whose potential term in the action is infinite and scales like $\frac{1}{a}$. In continuum language, their action will be proportional to a one-dimensional delta function $\delta(0)$. Naively, such configurations should be excluded. However, we will show that sometimes such configurations lead to states with energy of order $\frac{1}{a}$ of the kind mentioned above.

## 1.1 Exotic Global Symmetries

Our analysis of these systems will use as a guiding principle their exotic global symmetries. We will follow and extend the discussion of [10] and characterize the systems by the properties of their symmetries. We will start by analyzing global $U(1)$ symmetries. Later we will also consider $\mathbb{Z}_N$ symmetries.

An ordinary global $U(1)$ symmetry is associated with a conserved Noether current $(J_0, J^i)$ satisfying

$$\partial_0 J_0 = \partial_i J^i \,, \tag{1.1}$$

with $J_0$ its time component and $J^i$ the spatial component, which is a vector of the spatial rotation group. The conserved charge is an integral over all of space of the time component of the current

$$Q = \int_{\text{space}} J_0 \,. \tag{1.2}$$

The current conservation equation (1.1) guarantees that it is conserved.

We will generalize this symmetry in two ways.

First, we will be interested in situations where the conservation equation (1.1) has more than one spatial derivative in the right hand side. In the simplest nontrivial case, there are two derivatives and we will refer to the symmetry as a *dipole global symmetry*.[5]

Second, we will allow the time component of the current $J_0^I$ to be in a nontrivial representation $\mathbf{R}_{\text{time}}$ of the rotation group, with $I$ an $\mathbf{R}_{\text{time}}$ index. This rotation group can be the full continuous rotation group, or only a finite subgroup of it. Related to that, the spatial component of the current $J^K$ will be in an appropriate representation $\mathbf{R}_{\text{space}}$ of the rotation group, with $K$ an $\mathbf{R}_{\text{space}}$ index. Then, the conservation equation (1.1) takes the form

$$\partial_0 J_0^I = \partial_i \partial_j \cdots J^K f_K^{ij\cdots, I} \tag{1.3}$$

with $f_K^{ij\cdots, I}$ an invariant tensor coupling the indices. We will refer to such a current as $(\mathbf{R}_{\text{time}}, \mathbf{R}_{\text{space}})$.

---

[5]In the literature of dipole global symmetry, the charge is usually an integral of the current multiplied by a linear function of $x^i$. Such an expression is only allowed in $\mathbb{R}^{D,1}$, but not on more general manifolds. Our presentation of the dipole symmetry will be based on the local conservation equation (1.3), and is insensitive to the details of the manifold.

One interesting aspect of these exotic symmetries is that for special choices of the conservation equation (1.3), the conserved charge does not have to be integrated over the entire space. We can have

$$Q^I = \int_\Sigma J_0^I \qquad (1.4)$$

with $\Sigma$ a closed subspace, which can depend on the index $I$. For example, $\Sigma$ can be a line along the $x$ direction. Then, the charge $Q^I$ is a function of the remaining coordinates – $y$ in a $2+1$-dimensional system or $y$ and $z$ in a $3+1$-dimensional system. Furthermore, as we will see, sometimes the time component of the current satisfies another differential condition that restricts the dependence on these coordinates.

In our examples the global symmetry will be $U(1)$ (rather than $\mathbb{R}$) and therefore the charges are quantized. In that case, if the charge $Q^I$ is integrated over a subspace and it depends on the other coordinates, it must be discontinuous. We will see examples where the charge $Q^I$ is an integer changing from point to point and other examples where $Q^I$ is a linear combinations of delta functions with integer coefficients.

As we said above, such behavior of global symmetry charges poses a challenge to a description in terms of a continuum quantum field theory.

## 1.2 Naturalness and Robustness

An essential issue in every quantum field theory is whether it is natural. This notion has two different meanings, both of them related to the global symmetries of the system.

First, as is common in high-energy physics, we postulate a global symmetry (in the short-distance theory) $G_{UV}$ and demand that the UV Lagrangian includes all the terms that are compatible with the global symmetry with coefficients of order one. Terms that violate the global symmetry are "naturally" excluded. This notion of naturalness was articulated by 't Hooft [11].

All our systems here and in [1,2] are natural in this sense. Our Lagrangians respect their global symmetries and do not include more generic terms that violate them. However, as we will discuss in Section 1.3, we exclude from the Lagrangian certain high derivative terms that respect the symmetry and can affect the quantitative physical conclusions. They do not affect the qualitative behavior.

In real condensed matter systems the short-distance model has very few global internal symmetries. Typically, $G_{UV}$ is $\mathbb{Z}_2$, or $U(1)$, or trivial. Let us consider the extreme case of a trivial $G_{UV}$. In that case, the question of naturalness becomes a question of robustness. We assume that by tuning the short distance parameters, we find a low-energy theory with an emergent global symmetry $G_{IR}$. Then, we deform slightly the short distance parameters and ask whether the low-energy system still preserves the emergent symmetry $G_{IR}$. This is determined by an analysis of the operators in the low-energy theory and in particular, by those operators in the low-energy theory that violate $G_{IR}$.

If the low-energy theory has relevant operators that violate $G_{IR}$, then a generic deformation of the short-distance system induces them in the low-energy system and breaks the symmetry there. Then, some level of fine tuning will be needed in order to find a $G_{IR}$ symmetric system at low energies.

If, however, the low-energy system has no $G_{IR}$ violating relevant operator (or if it does not have any local operator at all), then a small deformation of the short-distance system does not ruin the symmetry at long distances. In that case the global symmetry of the low-energy system is robust. It is referred to as an emergent global symmetry or as an accidental global symmetry.[6]

---

[6]A known example in high-energy physics is the conservation of the global $U(1)$ symmetry of baryon minus

It should be emphasized that when we discuss the global symmetries $G_{UV}$ and $G_{IR}$ we should take into account all the global symmetries, including standard ones, the higher-form global symmetries of [9], and the more exotic ones of [10], this paper, and [1, 2]. Using such generalized global symmetries, it is clear why when a gauge symmetry emerges at long distances it is often robust.[7] See [12, 13] for early related discussions.

A weaker notion of robustness is also useful. We can start at short-distances with a non-trivial global symmetry $G_{UV}$. Then, the global symmetry of the long-distance theory $G_{IR}$ can have new symmetry elements that are not present in $G_{UV}$. Is this symmetry enhancement robust? The low-energy theory can have relevant operators violating $G_{UV}$. These operators are naturally absent in the Lagrangian, because we impose $G_{UV}$ at short distances. However, if the low-energy theory has relevant operators that preserve $G_{UV}$, but violate $G_{IR}$, then the enhanced global symmetry is not robust. Conversely, if there are no such relevant $G_{IR}$ violating operators, the symmetry enhancement is robust. To summarize, the robustness of the long-distance global symmetry $G_{IR}$ can depend on what the short-distance symmetry $G_{UV}$ is.

Not all of our models are robust. The long-distance theory might include relevant operators violating some global symmetries. However, some of our models, and in particular, the model of Section 7 and the models of [2] are both natural and robust. The latter ones are the low-energy field theories for the $3 + 1$-dimensional X-cube model.

*Examples*

Let us demonstrate this notion of robustness in some familiar examples.

In the $1+1$-dimensional XY-model, the microscopic symmetry $G_{UV}$ is the $U(1)$ momentum global symmetry. The long-distance field theory is based on a compact scalar field.[8] Its global symmetry $G_{IR}$ includes $G_{UV}$ and an emergent $U(1)$ winding global symmetry. If the radius of the scalar is large enough, the emergent winding symmetry is robust. However, if that radius is small, then the emergent winding symmetry is not robust and is generically broken by relevant winding operators. It is common and natural (in the technical sense) to impose that winding symmetry on the low-energy theory even for small radius. Then the theory is gapless for every radius and exhibits a duality, known in the string-theory literature as T-duality.

$1+1$-dimensional ordinary $U(1)$ gauge theory has a one-form global symmetry [9], which is referred to as electric. This global symmetry exists both in the lattice formulation of the system and in its continuum field theory. We can change the system by adding to it massive charged matter fields and then this one-form global symmetry is absent. However, this symmetry can emerge in the low-energy theory. In this case $G_{UV}$ is trivial and $G_{IR}$ is the electric one-form symmetry.

This $U(1)$ gauge theory is sometimes represented on the lattice in the Hamiltonian formalism by imposing Gauss law energetically, rather than as a constraint. (This was reviewed from the perspective we use here in [10].) Then, the lattice theory is not a gauge theory and one can deform it further by adding lattice operators violating the gauge symmetry, e.g. a term linear in the link variable. This lattice theory does not have the one-form global symmetry. However,

---

lepton number in the Standard Model. All the gauge invariant operators constructed out of the fields of the Standard Model that violate this symmetry are irrelevant. And therefore, this is an emergent global symmetry of the theory. In fact, all the renormalizable operators of the Standard Model preserve separately baryon number and lepton number. But since the corresponding global symmetries are violated by instantons of the weak interactions, we would not refer to these symmetries as accidental.

[7]We thank S. Shenker for many extremely stimulating discussions about this issue.

[8]Here and elsewhere in this paper we follow the high-energy physics terminology, where this means that the field is circle-valued, regardless of its action. This is to be contrasted with the notion of a compact field in the condensed-matter literature, which means that the lattice action does not preserve the winding symmetry.

it is easy to see that if this deformation is sufficiently small, its effect on the low-energy theory is negligible as the number of sites of the lattice $L$ goes to infinity. Therefore, even though the microscopic theory does not have the electric one-form symmetry, it emerges at long distances as a nontrivial $G_{IR}$ and it is robust.

This system should be contrasted with three variants of it.

First, consider the $1 + 1$-dimensional $\mathbb{Z}_N$ gauge theory. One way to construct this theory is by Higgsing the $U(1)$ gauge theory by a charge $N$ field. The presence of the charge $N$ field breaks the one-form $U(1)$ global symmetry to $\mathbb{Z}_N$ [9]. A convenient way to represent this theory is as a $BF$-theory with the Lagrangian [9, 14–16]

$$\frac{N}{2\pi}BdA \qquad (1.5)$$

with $B \sim B + 2\pi$ a circle-valued scalar field and $A$ is a one-form gauge field. Alternatively, we can formulate it directly as a $\mathbb{Z}_N$ gauge theory either on the lattice or in the continuum. In all of these formulations it has a robust electric $\mathbb{Z}_N$ one-form symmetry.

This system also has a magnetic symmetry, which is an ordinary (zero-form) $\mathbb{Z}_N$ global symmetry. The local operators charged under it are often called twist fields and in the formulation of the theory (1.5), it is represented by $e^{iB}$. The theory has $N$ ground states in which the expectation values of the twist field are $e^{2\pi in/N}$ with $n = 0, ..., N-1$. This means that (1.5) describes a spontaneously broken global $\mathbb{Z}_N$ symmetry.

If we start at short distances with an exact global $\mathbb{Z}_N$ magnetic symmetry (as in the Ising model), this symmetry remains exact at low energies. In the broken phase, the low-energy system can be described by (1.5). In this phase the one-form $\mathbb{Z}_N$ global symmetry emerges at long distances and it is robust.

Alternatively, if we start with the lattice $\mathbb{Z}_N$ gauge theory without the magnetic symmetry, then the microscopic symmetry $G_{UV}$ is only the electric one-form symmetry. We can still end up in a phase described by (1.5) with an emergent $\mathbb{Z}_N$ ordinary global symmetry. However, in this case the emergent global symmetry $G_{IR}$ is not robust. We can deform the low-energy theory (1.5) by $e^{iB}$ and explicitly break the $\mathbb{Z}_N$ magnetic global symmetry. This deformation lifts the degeneracy between the $N$ states.

As another variant of the $1 + 1$-dimensional $U(1)$ gauge theory, we can consider its $2 + 1$-dimensional version. Just like the $1+1$-dimensional system, it has an electric one-from global symmetry. This symmetry is robust. But unlike the $1 + 1$-dimensional $U(1)$ gauge theory, it has a magnetic $U(1)$ global symmetry, whose Noether current is $J^\mu = \frac{1}{2\pi}\epsilon^{\mu\nu\rho}\partial_\nu A_\rho$. This symmetry is not present in the lattice formulation of the system, and we could ask whether it is an emergent symmetry.

The simplest way to answer this question [17] is to dualize the gauge field to a free compact scalar $\varphi \sim \varphi + 2\pi$. Then the conserved magnetic current is $J^\mu \sim \partial^\mu\varphi$. The operators charged under this symmetry are known as monopole operators (see the modern discussion in [18]) and are represented by $e^{i\varphi}$. If the short-distance theory does not have this magnetic symmetry (i.e. $G_{UV}$ includes only the one-form global symmetry) and the system is not fine-tuned, then the long-distance theory is generically deformed by these monopole operators. This deformation breaks the magnetic symmetry and gaps the system. Therefore, the magnetic symmetry is not robust. This is the $2 + 1$-dimensional analog of the lack of robustness of the gapless $1 + 1$-dimensional XY-model at small radius.

Finally, we discuss a $2 + 1$-dimensional $\mathbb{Z}_N$ gauge theory. It has known lattice and continuum descriptions. One way to think about it in the continuum is by Higgsing the $U(1)$ theory. This description can be dualized to (1.5), this time with $B$ a $U(1)$ gauge field [9, 14–16]. The lattice system has an electric $\mathbb{Z}_N$ one-form symmetry $G_{UV}$. The symmetry $G_{IR}$ of the continuum theory further includes a $\mathbb{Z}_N$ one-form magnetic symmetry [9]. This emergent magnetic symmetry is different than in the $1+1$-dimensional $\mathbb{Z}_N$ gauge theory, where the magnetic symmetry

is an ordinary (zero-form) global symmetry. Correspondingly, the operators charged under it are not point-like. They are line operators $e^{i \oint B}$. Since they are not point-like, they cannot be added to the Lagrangian and therefore, the emergent magnetic symmetry is robust.

## 1.3 Discontinuous Field Configurations and Universality

One of the reasons to look for a low-energy effective theory is that it is universal. This universality usually arises as follows. The high-energy theory has many (in fact infinitely many) coupling constants, all of them affect the physical observables. When we focus on low-energy observables, most of the dependence on the microscopic parameters is irrelevant. The low-energy effective theory captures this fact as follows. It has a finite number of relevant and marginal operators, whose coefficients affect the low-energy observables. The coefficient of irrelevant operators are suppressed by the high-energy scale, which in our case is a power of the lattice spacing $a$. Therefore, their effect on low-energy observables is negligible. As a result, the low-energy effective theory allows us to organize the dependence of low-energy observables on the parameters more efficiently – more universally. It exhibits the relevant parameters and hides the irrelevant ones.

Let us demonstrate it in a standard Euclidean scalar field theory. The low-energy theory includes terms like $(\partial_\mu \phi)^2$. The effect of higher derivative terms in the Lagrangian on low momentum processes is negligible. They lead to corrections suppressed by powers of $a^2 k^2$ (where $k$ is a characteristic momentum). This is equivalent to saying that the effective action is a power series in derivatives and the higher derivative terms are irrelevant.

In the examples below we will see two notable exceptions to this general picture. First, in some cases, certain low derivative terms will be absent. This can happen either by gauge invariance or by imposing a global symmetry. Then, the most relevant term in the Lagrangian is a higher derivative term. Usually such higher derivative terms are negligible, but in this case they are not.

More important for us is the fact that in some of our examples the expansion in derivatives is not valid. This happens when some low-energy observables receive contributions associated with arbitrarily large momenta. This is another manifestation of the UV/IR mixing we mentioned above. In that case, we can compute the observable using the leading order terms in the Lagrangian, but terms with more derivatives lead to equally important contributions.

Let us discuss it in more detail and for concreteness focus again on the Euclidean scalar field theory. Starting at short distance with a lattice, the configurations in the functional integral are discontinuous. The values of the field $\phi$ at different lattice sites are independent. As we said, the continuum limit includes terms of the form $(\partial_\mu \phi)^2$, which force the field $\phi$ to be continuous. In some of our cases, such terms will be absent and we will be motivated to explore discontinuous field configurations.

For such discontinuous configurations, the expansion in the number of derivatives is questionable. Indeed, if we have a configuration $\phi$ with discontinuities in $x$, then the derivative $\partial_x \phi$ is infinite. And then terms in the Lagrangian with higher powers of $\partial_x$, which are nominally suppressed by powers of the lattice spacing $a$ (or some other UV cutoff), are not small. The effect of such terms on the various observables, might not be negligible and they might ruin the universality of the computation in the low-energy theory.

Below we will demonstrate these issues in various cases.[9] We will study continuum Lagrangians focusing on the leading order terms, and ignoring higher derivative terms. Then, we will discuss various discontinuous configurations, both with finite and with infinite action. We will show that in some cases higher derivative terms can change the quantitative results

---

[9]We thank P. Gorantla and H.T. Lam for helpful discussions about these points.

obtained by using only the leading terms. However, in the absence of fine tuning, they will not affect the qualitative features.

Finally, we relate this discussion to a comment we made at the beginning of Section 1.2 when we discussed naturalness. The higher derivative terms that we neglect are invariant under all the symmetries of the problem. The assumption of naturalness forces us to add them to the Lagrangian with appropriate coefficients. Since we neglected them, those models where these terms can change the quantitative conclusions are not strictly natural.

All these subtleties will not arise in Section 8, when we will discuss a gapped model. We will argue that in this case these higher derivative terms are indeed negligible. In fact, there is no local operator respecting the symmetry in the continuum Lagrangian, and the results obtained from that Lagrangian are universal.

## 1.4 Summary

In Section 2, we follow [19] and study a $2+1$-dimensional XY-model with interactions around plaquettes, the XY-plaquette model. After reviewing the lattice model, we discuss a continuum Lagrangian for the system. It is based on a compact (i.e. circle-valued) scalar field $\phi \sim \phi + 2\pi$ with the Lagrangian [19–25] (related Lagrangians appeared in [26–28])[10]

$$\mathcal{L} = \frac{\mu_0}{2}(\partial_0\phi)^2 - \frac{1}{2\mu}(\partial_x\partial_y\phi)^2. \tag{1.6}$$

It has two dipole global symmetries, which we refer to as momentum and winding (see Table 1). As mentioned before, discontinuous configurations of $\phi$ will play an important role in the analysis of this model. In particular, the discontinuities imply that we should also identify $\phi \sim \phi + 2\pi w^x(x) + 2\pi w^y(y)$ where $w^x(x), w^y(y) \in \mathbb{Z}$ are two discontinuous, integer-valued functions.

In Section 3, we study the continuum limit more carefully and explore the space of functions of our theory.

Section 4 is devoted to the spectrum of the model. Plane waves with generic momenta lead to a Fock space of states with energies of order one in the continuum limit. The states that are charged under the global symmetries have energy of order $\frac{1}{\ell a}$ with $\ell$ the physical length of the system ($\ell^x$ or $\ell^y$). In the standard continuum limit, $a$ is taken to zero with fixed $\ell$ and the energy of these states diverges. A conservative approach would simply discard these states. But being ambitious, we will analyze them in detail.[11]

In Section 5, we perform a duality transformation on (1.6) to find a theory of a compact field $\phi^{xy} \sim \phi^{xy} + 2\pi$ in the spin-two representation of the rotation group $\mathbb{Z}_4$ with the Lagrangian[12]

$$\mathcal{L} = \frac{\widetilde{\mu}_0}{2}(\partial_0\phi^{xy})^2 - \frac{1}{2\widetilde{\mu}}(\partial_x\partial_y\phi^{xy})^2$$

$$\widetilde{\mu}_0 = \frac{\mu}{4\pi^2}, \quad \widetilde{\mu} = 4\pi^2\mu_0. \tag{1.7}$$

The duality exchanges the momentum and winding symmetries between (1.6) and (1.7) (see Table 1).

Such an exchange of momentum and winding states is reminiscent of T-duality of a compact scalar in $1+1$ dimensions. This is particularly surprising in this $2+1$-dimensional system

---

[10]See [29] for a similar model. It also exhibits a subsystem symmetry and the related UV/IR mixing.

[11]One might also be interested in the limit $\ell \to \infty$ with fixed $a$. This is the large volume limit with fixed lattice spacing. In this limit, these states have low energy and must be included.

[12]Since the rotation group $\mathbb{Z}_4$ is finite and the system has a charge conjugation symmetry, we can combine a 90 degree rotation with charge conjugation and interpret $\phi^{xy}$ as having spin zero. Similarly, by combining parity with charge conjugation we can take it to be parity even.

Table 1: Global symmetries and their charges in the $2+1$-dimensional scalar theories $\phi$ and $\phi^{xy}$. The energies of states that are charged under these global symmetries are of order $1/a$. Detailed explanations will be given in the body of the paper.

| Lagrangian | $\frac{\mu_0}{2}(\partial_0\phi)^2 - \frac{1}{2\mu}(\partial_x\partial_y\phi)^2$ | $\frac{\tilde{\mu}_0}{2}(\partial_0\phi^{xy})^2 - \frac{1}{2\tilde{\mu}}(\partial_x\partial_y\phi^{xy})^2$ |
|---|---|---|
| dipole symmetry $(\mathbf{1}_0, \mathbf{1}_2)$ | momentum $(J_0 = \mu_0\partial_0\phi, J^{xy} = -\frac{1}{\mu}\partial^x\partial^y\phi)$ | winding $(J_0 = \frac{1}{2\pi}\partial_x\partial_y\phi^{xy}, J^{xy} = \frac{1}{2\pi}\partial_0\phi^{xy})$ |
| currents | $\partial_0 J_0 = \partial_x\partial_y J^{xy}$ | |
| charges | $Q^x(x) = \oint dy J_0 = \sum_\alpha N^x_\alpha \delta(x-x_\alpha)$ $Q^y(y) = \oint dx J_0 = \sum_\beta N^y_\beta \delta(y-y_\beta)$ $\oint dx Q^x(x) = \oint dy Q^y(y)$ | |
| energy | $\mathcal{O}(1/a)$ | |
| number of sectors | $L^x + L^y - 1$ | |
| dipole symmetry $(\mathbf{1}_2, \mathbf{1}_0)$ | winding $(J^{xy}_0 = \frac{1}{2\pi}\partial^x\partial^y\phi, J = \frac{1}{2\pi}\partial_0\phi)$ | momentum $(J^{xy}_0 = \tilde{\mu}_0\partial_0\phi^{xy}, J = -\frac{1}{\tilde{\mu}}\partial_x\partial_y\phi^{xy})$ |
| currents | $\partial_0 J^{xy}_0 = \partial^x\partial^y J$ | |
| charges | $Q^{xy}_x(x) = \oint dy J^{xy}_0 = \sum_\alpha W^x_\alpha \delta(x-x_\alpha)$ $Q^{xy}_y(y) = \oint dx J^{xy}_0 = \sum_\beta W^y_\beta \delta(y-y_\beta)$ $\oint dx Q^{xy}_x(x) = \oint dy Q^{xy}_y(y)$ | |
| energy | $\mathcal{O}(1/a)$ | |
| number of sectors | $L^x + L^y - 1$ | |
| duality map | $\mu_0 = \frac{\tilde{\mu}}{4\pi^2} \quad \mu = 4\pi^2\tilde{\mu}_0$ | |

Table 2: Analogy between the ordinary $1+1$-dimensional compact scalar theory and the $2+1$-dimensional $\phi$-theory. Detailed explanations will be given in the body of the paper.

|  | $(1+1)d$ compact scalar | $(2+1)d$ $\phi$ theory |
|---|---|---|
| lattice | XY-model | XY-plaquette model |
| fields | $\Phi \sim \Phi + 2\pi$ | $\phi \sim \phi + 2\pi w^x(x) + 2\pi w^y(y)$ |
| Lagrangian | $\mathcal{L} = \frac{R^2}{4\pi}(\partial_0 \Phi)^2 - \frac{R^2}{4\pi}(\partial_x \Phi)^2$ | $\mathcal{L} = \frac{\mu_0}{2}(\partial_0 \phi)^2 - \frac{1}{2\mu}(\partial_x \partial_y \phi)^2$ |
| global symmetry | momentum $\partial_0 J_0 = \partial_x J^x$ $(J_0 = \frac{R^2}{2\pi}\partial_0 \Phi, J^x = \frac{R^2}{2\pi}\partial^x \Phi)$ <br><br> winding $\partial_0 J_0^x = \partial^x J$ $(J_0^x = \frac{1}{2\pi}\partial^x \Phi, J = \frac{1}{2\pi}\partial_0 \Phi)$ | momentum dipole $\partial_0 J_0 = \partial_x \partial_y J^{xy}$ $(J_0 = \mu_0 \partial_0 \phi, J^{xy} = -\frac{1}{\mu}\partial^x \partial^y \phi)$ <br><br> winding dipole $\partial_0 J_0^{xy} = \partial_x \partial_y J^{xy}$ $(J_0^{xy} = \frac{1}{2\pi}\partial^x \partial^y \phi, J = \frac{1}{2\pi}\partial_0 \phi)$ |
| duality | T-duality $R \leftrightarrow 1/R$ | Self-duality $4\pi^2 \mu_0 \leftrightarrow \mu$ |

because all these states have energies of order $\frac{1}{a}$. The analogy between these $1+1$ and $2+1$-dimensional systems is summarized in Table 2.

Section 7 studies a gauge theory associated with the dipole global symmetry [20, 22, 23, 30–33]. (Related models were discussed in [25, 27, 28, 34–44].) In most of the papers about tensor gauge fields, the spatial components of the gauge field is a symmetric tensor. Here, we follow the global dipole symmetry above and have a single spatial field in the spin two of $\mathbb{Z}_4$ with the gauge transformation rule

$$
\begin{aligned}
A_0 &\to A_0 + \partial_0 \alpha, \\
A_{xy} &\to A_{xy} + \partial_x \partial_y \alpha
\end{aligned}
\tag{1.8}
$$

with $\alpha \sim \alpha + 2\pi$. There are no $A_{xx}$ and $A_{yy}$ components.

The gauge invariant electric field is

$$
E_{xy} = \partial_0 A_{xy} - \partial_x \partial_y A_0.
\tag{1.9}
$$

This gauge theory is similar to an ordinary $U(1)$ gauge theory in $1+1$ dimensions. It does not have a magnetic field and it has a $\theta$-parameter. The Lorentzian Lagrangian is

$$
\mathcal{L} = \frac{1}{g_e^2} E_{xy}^2 + \frac{\theta}{2\pi} E_{xy}.
\tag{1.10}
$$

Table 3: Analogy between the $1+1$-dimensional $U(1)$ gauge theory and the $2+1$-dimensional $U(1)$ tensor gauge theory. Detailed explanations will be given in the body of the paper.

|  | $(1+1)d\ U(1)$ gauge theory | $(2+1)d\ U(1)$ tensor gauge theory |
|---|---|---|
| gauge fields | $A_\mu \sim A_\mu + \partial_\mu \alpha$ | $A_0 \sim A_0 + \partial_0 \alpha$ <br> $A_{xy} \sim A_{xy} + \partial_x \partial_y \alpha$ |
| field strengths | $E_x = \partial_0 A_x - \partial_x A_0$ | $E_{xy} = \partial_0 A_{xy} - \partial_x \partial_y A_0$ |
| flux | $\oint d\tau \oint dx\, E_x = 2\pi n$ | $\oint d\tau \oint dy\, E_{xy} = 2\pi \sum_\alpha n_{x\alpha} \delta(x - x_\alpha)$ <br> $\oint d\tau \oint dx\, E_{xy} = 2\pi \sum_\beta n_{y\beta} \delta(y - y_\beta)$ |
| Lagrangian | $\mathcal{L} = \frac{1}{g^2} E_x^2 + \frac{\theta}{2\pi} E_x$ | $\mathcal{L} = \frac{1}{g_e^2} E_{xy}^2 + \frac{\theta}{2\pi} E_{xy}$ |
| EoM | $\partial_0 E_x = 0$ | $\partial_0 E_{xy} = 0$ |
| Gauss law | $\partial_x E_x = 0$ | $\partial_x \partial_y E_{xy} = 0$ |
| $U(1)$ global symmetry | electric one-form <br> $\partial_0 J_0^x = 0,\ \ \partial_x J_0^x = 0$ <br> $J_0^x = \frac{2}{g^2} E_x + \frac{\theta}{2\pi}$ | electric tensor <br> $\partial_0 J_0^{xy} = 0,\ \ \partial_x \partial_y J_0^{xy} = 0$ <br> $J_0^{xy} = \frac{2}{g_e^2} E_{xy} + \frac{\theta}{2\pi}$ |

Just as an ordinary $1+1$-dimensional $U(1)$ gauge theory is effectively a quantum mechanical system of a single variable, the holonomy, this system is also lower dimensional. It is effectively $1+1$-dimensional. In particular, it has no local excitation in $2+1$ dimensions. The effective $1+1$-dimensional system is quite peculiar. The energy of its states is of order $\ell a$. This is to be contrasted with the charged states of the non-gauge theory (Section 4), whose energy is of order $\frac{1}{\ell a}$. In the standard continuum limit, $a \to 0$ with fixed $\ell$, the energy of these states goes to zero and the system has an infinite vacuum degeneracy.[13] The analogy between these $1+1$ and $2+1$-dimensional systems is summarized in Table 3.

In Section 8, we consider a $\mathbb{Z}_N$ version of this $U(1)$ tensor gauge theory. We present two dual continuum Lagrangians of this system. First, we Higgs the $U(1)$ gauge theory to $\mathbb{Z}_N$ using a scalar $\phi$. Then, we dualize $\phi$, as in Section 5, to $\phi^{xy}$ and find a $BF$-type description of the system.

---

[13]Alternatively, as in the non-gauge theory, we can consider the large volume limit, $\ell \to \infty$ with fixed $a$, and then the energy of these states diverges. This signals the fact that the spectrum of local excitations is gapped.

The spectrum of this theory has $N^{L^x+L^y-1}$ states. It is infinite in the continuum limit. Its entropy scales like the length of the system.

We also present two lattice models that lead at long distances to this continuum model. One of them is based on a $\mathbb{Z}_N$ tensor gauge theory and the other is a $\mathbb{Z}_N$ version of the lattice XY-plaquette model of Section 2, also known as the plaquette Ising model (see [45] for a review and earlier references).

Table 4: Comparison between the ordinary $1 + 1$-dimensional $\mathbb{Z}_N$ gauge theory and the $2 + 1$-dimensional $\mathbb{Z}_N$ tensor gauge theory. Detailed explanations will be given in the body of the paper.

| | $(1+1)d$ $\mathbb{Z}_N$ gauge theory | $(2+1)d$ $\mathbb{Z}_N$ tensor gauge theory |
|---|---|---|
| lattice | $\mathbb{Z}_N$ lattice gauge theory or $\mathbb{Z}_N$ Ising model | $\mathbb{Z}_N$ lattice tensor gauge theory or $\mathbb{Z}_N$ plaquettte Ising model |
| fields | $B \sim B + 2\pi$ <br> $A_\mu \sim A_\mu + \partial_\mu \alpha$ | $\phi^{xy} \sim \phi^{xy} + 2\pi w^x(x) + 2\pi w^y(y)$ <br> $A_0 \sim A_0 + \partial_0 \alpha$ <br> $A_{xy} \sim A_{xy} + \partial_x \partial_y \alpha$ |
| Lagrangian | $\mathcal{L} = \frac{N}{2\pi} B E_x$ | $\mathcal{L} = \frac{N}{2\pi} \phi^{xy} E_{xy}$ |
| $\mathbb{Z}_N$ global symmetry & symmetry operator | electric one-form <br> $\exp[iB]$ <br><br> ordinary zero-form <br> $\exp[i \oint dx A_x]$ | electric tensor <br> $\exp[i\phi^{xy}]$ <br><br> dipole <br> $\exp[i \int_{x_1}^{x_2} dx \oint dy A_{xy}]$ <br> $\exp[i \oint dx \int_{y_1}^{y_2} dy A_{xy}]$ |
| defect | $\mathbb{Z}_N$ charged probe particles <br><br> $\exp[i \int_{\mathcal{C}} (dt A_0 + dx A_x)]$ | fractons <br><br> $\exp[i \int_{-\infty}^{\infty} dt A_0]$ <br> $\exp[i \int_{x_1}^{x_2} dx \int_{\mathcal{C}} (dt \partial_x A_0 + dy A_{xy})]$ |
| ground state degeneracy on a torus | $N$ | $N^{L^x+L^y-1}$ |

As in the examples in Table 2 and Table 3, this $2 + 1$-dimensional $\mathbb{Z}_N$ tensor gauge theory is analogous to a $1 + 1$-dimensional ordinary $\mathbb{Z}_N$ gauge theory. We summarize this analogy in Table 4. It is nice to see the hierarchy between these three situations. The systems in Table 2 have gapless local excitations. The systems in Table 3 do not have local excitations. Their

Table 5: Spectra of the continuum field theories discussed in this paper. Depending on the order of limits $a \to 0$ or $\ell \to \infty$, the energy of the charged states goes to zero or infinity.

| $(2+1)d$ | Lagrangian | spectrum |
|---|---|---|
| scalar theory $\phi$ | $\frac{\mu_0}{2}(\partial_0 \phi)^2 - \frac{1}{2\mu}(\partial_x \partial_y \phi)^2$ | gapless local excitations charged states at order $\frac{1}{\mu \ell a}$, $\frac{1}{\mu_0 \ell a}$ |
| $U(1)$ tensor gauge theory $A$ | $\frac{1}{g_e^2}E_{xy}^2 + \frac{\theta}{2\pi}E_{xy}$ | no local excitations – gapped charged states at order $g_e^2 \ell a$ |
| $\mathbb{Z}_N$ tensor gauge theory | $\frac{N}{2\pi}\phi^{xy}E_{xy}$ | no local excitations – gapped large vacuum degeneracy |

excitations behave like those of a lower dimensional theory. And the systems in Table 4 have only a finite number of states (which diverges as $L^i \to \infty$). We compare these theories in Table 5.

Finally, in Appendix A, we compute correlation functions of the XY-plaquette model in the continuum limit and demonstrate the subtleties in the space of functions.

Throughout this paper our spacetime will be flat. Space will be either a plane $\mathbb{R}^2$ or a two-torus $\mathbb{T}^2$. The signature will be either Lorentzian or Euclidean. And when it is Euclidean we will also consider spacetime to be a three-torus $\mathbb{T}^3$. We will use $x^i$ with $i = 1, 2$ to denote the two spatial coordinates, $x^0$ to denote Lorentzian time, and $\tau$ for the Euclidean time. The spatial vector index $i$ can be freely raised and lowered. When specializing to a particular component of an equation, we will also use $(t, x, y)$ to denote the coordinates with $t \equiv x^0, x \equiv x^1, y \equiv x^2$. When we will consider tensors, e.g. $A_{ij}$, we will denote specific components also as $A_{xy}$.

## 1.5 Preview of [1, 2]

We will continue this line of investigation in [1] and [2], where we will present 3+1-dimensional versions of the systems in this paper. Just as the examples here are analogous to certain ordinary $1 + 1$-dimensional systems, there will be analogies between the examples in [1] and [2] and certain ordinary $2 + 1$-dimensional systems.

In [1] we will discuss two non-gauge systems. The first is an XY-plaquette model, which is described at long distances by a scalar field $\phi$. The discussion will be quite similar to the analysis in this paper. The other non-gauge theory will be based on a dynamical field $\hat{\phi}$ in the tensor representation of the rotation group. More explicitly, we will limit ourselves to systems whose rotation symmetry is the cubic group (just as we limit ourselves here to systems with a $\mathbb{Z}_4$ rotation symmetry) and the dynamical field $\hat{\phi}$ will be in the two-dimensional representation of that group. As in this paper, we will find exotic momentum and winding global symmetries and will explore the spectrum of states charged under these global symmetries.

We will then study two different $U(1)$ gauge theories obtained by gauging the momentum

symmetries of the $\phi$ and $\hat{\phi}$ systems. We will denote the gauge fields by $A$ and $\hat{A}$, respectively. Certain aspects of the gauge theory of $A$ have been discussed in [20,21,24,30–32] (see [22,23, 25,27,28,33–44] for related tensor gauge theories). The gauge theory of $\hat{A}$ is related to gauge theories discussed in [20]. These two gauge theories have new exotic global symmetries, analogous to the electric and the magnetic generalized global symmetries of ordinary $U(1)$ gauge theories [9] and [10]. And they have subtle excitations carrying these global electric and magnetic charges. (This is similar to what we see in Section 7.)

We will also show that the non-gauge theory of $\phi$ is dual to the $\hat{A}$ gauge theory. Similarly, the non-gauge theory of $\hat{\phi}$ is dual to the $A$ gauge theory. In every one of these dual pairs the global symmetries and the spectra match across the duality. This is particularly surprising given the subtle nature of the states that are charged under the momentum and winding symmetries of the non-gauge systems and the magnetic and electric symmetries of the gauge systems.

In [2] we will study a $3 + 1$-dimensional version of the discussion in Section 8. We will present three dual continuum Lagrangians of the $\mathbb{Z}_N$ tensor gauge theory. (One of these was discussed in [20].) We will analyze the global symmetries, the gauge invariant observables/defects, and the spectrum of the Hamiltonian. We will match these continuum models with three different lattice theories, which are dual at long distances. One of these lattice models is the celebrated X-cube model [7].

## 2 The XY-Plaquette Model

### 2.1 The Lattice Model

In this section we review the $2 + 1$-dimensional XY-plaquette model and its analysis in [19].

We study the system on a spatial lattice with $L^x$ and $L^y$ sites in the $x$ and $y$ directions and we use periodic boundary conditions. We label the sites by $s = (\hat{x}, \hat{y})$, with integer $\hat{x} = 1, \cdots, L^x$ and $\hat{y} = 1, \cdots, L^y$. When we later take the continuum limit, we will use $x = a\hat{x}$ and $y = a\hat{y}$ to label the coordinates and $\ell^x = aL^x$ and $\ell^y = aL^y$ to denote the physical size of the system.

The degrees of freedom are phase variable $e^{i\phi_s}$ (and therefore $\phi_s \sim \phi_s + 2\pi$). Their conjugate momenta $\pi_s$ satisfy

$$[\phi_s, \pi_{s'}] = i\delta_{s,s'}. \tag{2.1}$$

The $2\pi$-periodicity of $\phi_s$ implies that the eigenvalues of $\pi_s$ are integers. The Hamiltonian is

$$H = \frac{u}{2} \sum_s (\pi_s)^2 - K \sum_s \cos(\Delta_{xy}\phi_s)$$

$$\Delta_{xy}\phi_{\hat{x},\hat{y}} = \phi_{\hat{x}+1,\hat{y}+1} - \phi_{\hat{x}+1,\hat{y}} - \phi_{\hat{x},\hat{y}+1} + \phi_{\hat{x},\hat{y}}. \tag{2.2}$$

This lattice system has a large number of $U(1)$ global symmetries, which grows linearly in the size of the system. For every point $\hat{x}_0$ in the $x$ direction, there is a $U(1)$ global symmetry that rotates the $\phi_s$'s on the $\hat{x}_0$-column simultaneously:

$$U(1)_{\hat{x}_0}: \quad \phi_s \to \phi_s + \varphi, \quad \forall s = (\hat{x}, \hat{y}) \text{ with } \hat{x} = \hat{x}_0, \tag{2.3}$$

where $\varphi \in [0, 2\pi)$. Similarly, for every point $\hat{y}_0$ in the $y$ direction, there is a $U(1)$ global symmetry that rotates the $\phi_s$'s on the $\hat{y}_0$-row simultaneously:

$$U(1)_{\hat{y}_0}: \quad \phi_s \to \phi_s + \varphi, \quad \forall s = (\hat{x}, \hat{y}) \text{ with } \hat{y} = \hat{y}_0. \tag{2.4}$$

There is one relation among these symmetries; the composition of all the $U(1)_{\hat{x}_0}$ is the same as the composition of all the $U(1)_{\hat{y}_0}$. It rotates all the $\phi_s$'s on the two-dimensional lattice

simultaneously. In total, we have $L^x + L^y - 1$ independent $U(1)$ global symmetries. We will refer to these symmetries as the momentum dipole symmetries.

Let us modify the XY-plaquette model by including the following interaction across two links:

$$\sum_{\hat{x},\hat{y}} e^{i\phi_{\hat{x}+1,\hat{y}}} e^{-2i\phi_{\hat{x},\hat{y}}} e^{i\phi_{\hat{x}-1,\hat{y}}} + e^{i\phi_{\hat{x},\hat{y}+1}} e^{-2i\phi_{\hat{x},\hat{y}}} e^{i\phi_{\hat{x},\hat{y}-1}} + c.c.. \tag{2.5}$$

This model and its global symmetries have been studied in [46]. This modification breaks most of the $U(1)$ global symmetries (2.3),(2.4), leaving only the overall $U(1)$ symmetry that rotates all the $\phi_s$'s by the same phase. If space is noncompact, there are also two additional $U(1)$ global dipole symmetries:

$$\begin{aligned}
U(1)'_x : & \quad \phi_s \rightarrow \phi_s + \hat{x}\,\varphi_x, \quad \forall s = (\hat{x}, \hat{y}), \\
U(1)'_y : & \quad \phi_s \rightarrow \phi_s + \hat{y}\,\varphi_y, \quad \forall s = (\hat{x}, \hat{y}).
\end{aligned} \tag{2.6}$$

with parameters $\varphi_x, \varphi_y \in [0, 2\pi)$. These symmetries are absent if we add also a more generic interaction of spins across a single link. We will discuss these symmetries in more detail when we consider this model in continuum following (2.15).

In Section 8.4 we will discuss the $\mathbb{Z}_N$ generalization of the XY-plaquette model.

## 2.2 First Attempt at a Continuum Theory

Here we present a first attempt for the continuum Lagrangian of the XY-plaquette model. Certain aspects of this field theory have been discussed in [19–25]. We will return to the more subtle issues in Section 3.

Let $\phi$ be a $2\pi$-periodic real scalar field. (See Section 3 for more details about its periodicity.) The Lagrangian in Lorentzian signature is

$$\mathcal{L} = \frac{\mu_0}{2} (\partial_0 \phi)^2 - \frac{1}{2\mu} (\partial_x \partial_y \phi)^2, \tag{2.7}$$

where $\mu_0, \mu$ are two parameters with mass dimension +1.

The equation of motion

$$\mu_0 \partial_0^2 \phi + \frac{1}{\mu} \partial_x^2 \partial_y^2 \phi = 0 \tag{2.8}$$

implies a *momentum dipole global symmetry* with currents [23]:

$$\begin{aligned}
J_0 = \mu_0 \partial_0 \phi, \quad J^{xy} = -\frac{1}{\mu} \partial^x \partial^y \phi, \\
\partial_0 J_0 = \partial_x \partial_y J^{xy}.
\end{aligned} \tag{2.9}$$

The $\mathbb{Z}_4$ representations of the currents are $(\mathbf{R}_{\text{time}}, \mathbf{R}_{\text{space}}) = (\mathbf{1}_0, \mathbf{1}_2)$.

Naively, we can write $J^{xy}$ as a total derivative of $\partial^i \phi$, and study a more elementary current with three spatial derivatives in the conservation equation. However, in Section 3, we will argue that $\partial^i \phi$ is not a well-defined operator in the continuum limit, while $\partial^x \partial^y \phi$ is.

The conserved charges of the momentum dipole symmetry are

$$\begin{aligned}
Q^x(x) = \oint dy\, J_0, \quad Q^y(y) = \oint dx\, J_0, \\
\oint dx\, Q^x(x) = \oint dy\, Q^y(y).
\end{aligned} \tag{2.10}$$

The momentum dipole symmetry shifts the scalar field by arbitrary functions of one spatial coordinate:

$$\phi(t,x,y) \to \phi(t,x,y) + c^x(x) + c^y(y). \qquad (2.11)$$

On a lattice with $L^i$ sites along the $x^i$ direction, the number of these charges is $L^x + L^y - 1$. These are the continuum limits of the $U(1)_{\hat{x}_0}, U(1)_{\hat{y}_0}$ symmetries on the lattice in Section 2.1.

When the global form of the momentum dipole symmetry is $U(1)$ (as opposed to $\mathbb{R}$), the charges $\int_{x_1^i}^{x_2^i} dx^i Q^i(x^i) \in \mathbb{Z}$ are integers at *any* interval $[x_1^i, x_2^i]$. Equivalently, the charges $Q^i(x^i)$ are linear combinations of delta functions with integer coefficients (see Section 4.1).

The continuum limit of the ordinary XY-model also has a winding symmetry (which is not present on the lattice):

$$J_0^i = \frac{1}{2\pi} \partial^i \phi, \quad J = \frac{1}{2\pi} \partial_0 \phi,$$
$$\partial_0 J_0^i = \partial^i J. \qquad (2.12)$$

As we will see in Section 3, the operator $\partial^i \phi$ is ill-defined in the XY-plaquette model, and hence, the ordinary winding symmetry (2.12) does not exist in its continuum limit.

Even though the ordinary winding symmetry (2.12) does not exist, there is a *winding dipole global symmetry* [23]:

$$J_0^{xy} = \frac{1}{2\pi} \partial^x \partial^y \phi, \quad J = \frac{1}{2\pi} \partial_0 \phi,$$
$$\partial_0 J_0^{xy} = \partial^x \partial^y J. \qquad (2.13)$$

The $\mathbb{Z}_4$ representations of the currents are $(\mathbf{R}_{\text{time}}, \mathbf{R}_{\text{space}}) = (\mathbf{1}_2, \mathbf{1}_0)$. This winding dipole symmetry cannot be integrated to (2.12) since $\partial^i \phi$ is not a well-defined operator. The winding dipole symmetry is a global symmetry of the continuum field theory, but not of the lattice XY-plaquette model.

The charges of the winding dipole symmetry are

$$Q_x^{xy}(x) = \oint dy\, J_0^{xy}, \quad Q_y^{xy}(y) = \oint dx\, J_0^{xy},$$
$$\oint dx\, Q_x^{xy}(x) = \oint dy\, Q_y^{xy}(y). \qquad (2.14)$$

Again, when the global form of the winding dipole symmetry is $U(1)$ (as opposed to $\mathbb{R}$), the charges $Q_i^{xy}(x^i)$ are linear combinations of delta functions with integer coefficients (see Section 3.2). On a lattice with $L^i$ sites along the $x^i$ direction, the number of these charges is $L^x + L^y - 1$.

Let us deform the Lagrangian (2.7) to:

$$\mathcal{L} = \frac{\mu_0}{2}(\partial_0 \phi)^2 - \frac{1}{2\mu}(\partial_x \partial_y \phi)^2 - \frac{1}{2\mu'}\left[(\partial_x^2 \phi)^2 + (\partial_y^2 \phi)^2\right]. \qquad (2.15)$$

The last term is the continuum limit of (2.5). In the special case when $\mu' = 2\mu$, the Lagrangian is invariant under the full continuous $SO(2)$ rotation group and not only its $\mathbb{Z}_4$ subgroup. (This $SO(2)$ invariant model was studied in [27].) This deformation renders the continuum limit of $\partial_i \phi$ smooth. Hence, the ordinary winding symmetry (2.12) exists and the winding dipole symmetry (2.13) becomes trivial.

The equation of motion of (2.15)

$$\mu_0 \partial_0^2 \phi + \frac{1}{\mu} \partial_x^2 \partial_y^2 \phi + \frac{1}{\mu'}(\partial_x^4 \phi + \partial_y^4 \phi) = 0 \qquad (2.16)$$

means that the conserved momentum dipole current (2.9) is modified

$$J_0 = \mu_0 \partial_0 \phi\,,$$
$$J^{xy} = -\frac{1}{\mu}\partial^x\partial^y\phi\,, \quad J^{xx} = -\frac{1}{\mu'}\partial^x\partial^x\phi\,, \quad J^{yy} = -\frac{1}{\mu'}\partial^y\partial^y\phi\,, \tag{2.17}$$
$$\partial_0 J_0 = \partial_x\partial_y J^{xy} + \partial_x\partial_x J^{xx} + \partial_y\partial_y J^{yy}\,.$$

Now the $\mathbb{Z}_4$ representations of the components are $(\mathbf{R}_{\text{time}}, \mathbf{R}_{\text{space}}) = (\mathbf{1}_0, \mathbf{1}_0 \oplus \mathbf{1}_2 \oplus \mathbf{1}_2)$. (Since $\partial_i\phi$ is a meaningful operator, we can also express it in terms of currents with a single derivative and a conservation equation with three derivatives.)

The previously conserved charges (2.10) are no longer conserved. But the overall $U(1)$ charge

$$Q = \oint dx\,dy\, J_0 \tag{2.18}$$

is still conserved. In addition, on $\mathbb{R}^2$ we also have two conserved dipole charges, which are integrated versions of (2.10) with a linear function of $x^i$:

$$q^x = \oint dx\,dy\, x\, J_0\,,$$
$$q^y = \oint dx\,dy\, y\, J_0\,. \tag{2.19}$$

They implement

$$\phi(t,x,y) \to \phi(t,x,y) + C^x x + C^y y\,, \tag{2.20}$$

with constants $C^x, C^y \in \mathbb{R}$. These dipole charges were studied in [27].

Note that although these dipole charges $q^x, q^y$ are meaningful only on $\mathbb{R}^2$, the conserved dipole current (2.17) is defined locally and it exists more generally.

# 3 The Fields

## 3.1 The Continuum Limit

In this section we carefully take the continuum limit of the XY-plaquette model.

As a warmup, let us start with the continuum limit of the XY-model. There is a phase variable $e^{i\phi_s}$ at every site $s = (\hat{x}, \hat{y})$. The interactions consists of terms across a link:

$$\exp[i\Delta_x\phi_s] \equiv \exp[i\phi_{\hat{x}+1,\hat{y}} - i\phi_{\hat{x},\hat{y}}]\,,$$
$$\exp[i\Delta_y\phi_s] \equiv \exp[i\phi_{\hat{x},\hat{y}+1} - i\phi_{\hat{x},\hat{y}}]\,. \tag{3.1}$$

For a typical lattice configuration, the difference between two neighboring $\phi_s$ is order 1, i.e. $\Delta_i\phi_s \sim 1$. In the continuum limit, we consider smooth configurations, such that

$$\Delta_i\phi_s \sim \frac{a}{\ell} \ll 1. \tag{3.2}$$

Here $a$ is the lattice spacing and $\ell$ is a characteristic size of the system. Since $\Delta_i\phi_s \ll 1$, it is clear how to resolve the ambiguity in assigning a real (as opposed to a circle-valued) $\Delta_i\phi_s$ to the link - take $|\Delta_i\phi_s| < \pi$. Then we can define the derivatives

$$\partial_i\phi_s = -ie^{-i\phi_s}\partial_i e^{i\phi_s} \equiv \frac{1}{a}\Delta_i\phi_s \sim \frac{1}{\ell}\,. \tag{3.3}$$

This definition makes sense even when $\phi_s$ is discontinuous as a real number, but is smooth as a phase, i.e. $e^{i\phi_s}$ varies slowly. By contrast, a typical lattice configuration has $\partial_i \phi_s \sim \frac{1}{a}$, which is excluded from the continuum theory.

Equivalently, we cover the space with patches in which $\phi_s$ is a real number and the transition functions involve shifts of $\phi_s$ by $2\pi\mathbb{Z}$. Then, the derivatives are taken in each patch.

Now we turn to the XY-plaquette model. We have a phase variable $e^{i\phi_s}$ at every site $s$ and the interactions are around plaquettes:

$$\exp[i\Delta_{xy}\phi_s] \equiv \exp\left[i(\phi_{\hat{x}+1,\hat{y}+1} - \phi_{\hat{x}+1,\hat{y}} - \phi_{\hat{x},\hat{y}+1} + \phi_{\hat{x},\hat{y}})\right]. \tag{3.4}$$

For a typical lattice configuration, the difference between two neighboring $\phi_s$'s is order 1, and therefore $\Delta_{xy}\phi_s \sim 1$. In the strict continuum limit, we consider smooth configurations, such that

$$\Delta_{xy}\phi_s \sim \frac{a^2}{\ell^2} \ll 1 \tag{3.5}$$

and we define the double derivative as

$$\partial_x \partial_y \phi_s \equiv \frac{1}{a^2}\Delta_{xy}\phi_s. \tag{3.6}$$

Again, since $\Delta_{xy}\phi_s \sim \frac{a^2}{\ell^2} \ll 1$, it is clear how to resolve the ambiguity in assigning a real (as opposed to a circle-valued) $\Delta_{xy}\phi_s$ - take $|\Delta_{xy}\phi_s| < \pi$. By contrast, a typical lattice configuration has $\partial_x \partial_y \phi_s \sim \frac{1}{a^2}$, which is excluded from the continuum theory.

One new element here, which was not present in the previous case of the XY-model, is that even if $\Delta_{xy}\phi_s \sim \frac{a^2}{\ell^2}$, $e^{i\phi_s}$ might not vary smoothly. For example, a configuration of $\phi_s$, which depends discontinuously only on one coordinate, say $x$, has $\Delta_{xy}\phi_s = 0$. Since the changes in $\phi_s$ between neighboring sites are not small, there is no natural way to define the derivative $\partial_x \phi_s$.

More precisely, we could try to define the derivative using the product across a link

$$\exp[ia\partial_x \phi_s] \equiv \exp[i\phi_{(\hat{x}+1,\hat{y})}]\exp[-i\phi_{(\hat{x},\hat{y})}], \tag{3.7}$$

but there is no natural way to define $\partial_x \phi_s$ without an additive $\frac{2\pi}{a}\mathbb{Z}$ ambiguity. Therefore, while the double derivative $\partial_x \partial_y \phi_s$ has a smooth continuum limit, the single derivative $\partial_i \phi_s$ does not.[14]

We can be more ambitious and study also more singular configurations. In the strict continuum limit, $\Delta_{xy}\phi_s \sim \frac{a^2}{\ell^2}$, but we can also consider configurations with

$$\Delta_{xy}\phi_s \sim \frac{a}{\ell} \ll 1. \tag{3.8}$$

Here $\partial_x \partial_y \phi$ includes terms of order $1/a$, or equivalently, a single delta function. Again, since $\Delta_{xy}\phi_s \sim \frac{a}{\ell} \ll 1$, it is clear how to resolve the ambiguity in assigning a real (as opposed to a circle-valued) $\Delta_{xy}\phi_s$ - take $|\Delta_{xy}\phi_s| < \pi$.

Let us consider a configuration whose $\Delta_{xy}\phi_s \sim a/\ell$:

$$\phi(x,y) = 2\pi \frac{x}{\ell^x} W^y(y), \tag{3.9}$$

where $W^y(y) \in \mathbb{Z}$ is an integer-valued, discontinuous function. We take it to be piecewise constant with a finite number of segments. $\phi(x,y)$ is continued periodically outside

---

[14]Because of this, we cannot write the double derivative as $-i\partial_x(e^{-i\phi_s}\partial_y e^{i\phi_s})$.

$0 \leq x < \ell^x, 0 \leq y < \ell^y$. Its second derivative $\partial_x \partial_y \phi = 2\pi \frac{1}{\ell^x} \partial_y W^y(y)$ involves the infinite function

$$\partial_y W^y(y) \equiv \frac{2\pi}{a} [W^y(y+a) - W^y(y)] \sim \frac{1}{a}. \tag{3.10}$$

Note that $\Delta_{xy}\phi_s = a^2 \partial_x \partial_y \phi \sim \frac{a}{\ell^x} \ll 1$. The action of this configuration is of order $1/a$.

When $W^y(y)$ is not piecewise constant, the action of this configuration can be higher than order $1/a$. Suppose the number of segments of $W^y(y)$ is of order $1/a$, such a configuration has action of order $1/a^2$. Even though its action is of the same order as a typical lattice fluctuation, it carries nontrivial charges under a winding global symmetry (see Section 4.2).

In summary, as long as $|\Delta_{xy}\phi_s| \ll 1$ (either $\Delta_{xy}\phi_s \sim \frac{a^2}{\ell^2}$ or $\Delta_{xy}\phi_s \sim \frac{a}{\ell}$), we can unambiguously resolve the $2\pi$-periodicity of $\Delta_{xy}\phi_s$ by assigning to it a real value $|\Delta_{xy}\phi_s| < \pi$ and then we can define the double derivative as (3.6). In the strict continuum limit, $\Delta_{xy}\phi_s \sim \frac{a^2}{\ell^2}$ and this second derivative is finite. For $\Delta_{xy}\phi_s \sim \frac{a}{\ell}$, it is infinite and scales like a one-dimensional delta function.

As discussed in Section 1.3, higher derivative terms and terms with higher powers of $\phi$ could potentially ruin the universality of the answers we get for such configurations. Indeed, the precise coefficient for the action of order $1/a$ can be modified by these terms. But these higher derivative terms generically do not change the qualitative features of these configurations.

We review the various classes of functions in the continuum limit of the XY-plaquette model:

- Typical lattice configurations have $\Delta_{xy}\phi_s \sim 1$. Since in the continuum limit, we take the lattice parameter $K$ in (2.2) to infinity, they are suppressed in the continuum limit.

- The configurations in the strict continuum limit have $\Delta_{xy}\phi_s \sim \frac{a^2}{\ell^2}$ and hence have a finite action. The corresponding states have finite energy. This set of configurations includes discontinuous functions.

- We are also interested in configurations with $\Delta_{xy}\phi_s \sim \frac{a}{\ell}$. Even though their action is of order $\frac{\ell}{a}$, which is infinite in the continuum limit, it is smaller than the action of the typical lattice configuration. Therefore, these configurations are distinct from the generic lattice configurations. We will discuss this in more detail below.

## 3.2 Transition Functions

Above, we discussed the continuum limit of the XY-plaquette model and found that while the continuum limit $\partial_x \partial_y \phi$ exists, that of $\partial_i \phi$ does not. In this section we will give another interpretation of these features from the continuum field theory point of view.

On the lattice, the fundamental variables are $U(1)$ phases $e^{i\phi_s}$. In the continuum, it is more natural to work with the $\mathbb{R}$-valued field $\phi$. The field $\phi$ is subject to discrete gauge symmetry and requires transition functions.

Locally, the real scalar $\phi$ is subject to the following discrete gauge transformation

$$\phi(t, x, y) \sim \phi(t, x, y) + 2\pi w^x(x) + 2\pi w^y(y), \tag{3.11}$$

where $w^i(x^i) \in \mathbb{Z}$ is any integer-valued, discontinuous function. In other words, we gauge a $\mathbb{Z}$ momentum dipole symmetry, so that the momentum dipole global symmetry is $U(1)$ (as opposed to $\mathbb{R}$). Because of this identification, operators such as $\partial_0 \phi, \partial_x \partial_y \phi, e^{i\phi}$ are well-defined local operators, while $\phi$ and $\partial^i \phi$ are not.

Let us discuss some global issues. In standard situations, $\phi$ is a smooth function. Then, the global structure is described using patches in which $\phi$ is a smooth real function and there are transition functions in the overlap regions between patches. Let us try to imitate such a

construction for our $\phi$, which is not smooth. A more detailed discussion will be presented in [47].

We cover the manifold with patches $\mathcal{U}_a$ that are locally open rectangles. $\phi_{(a)}$ is single-valued in each rectangle $\mathcal{U}_a$. At the overlap between two patches $\mathcal{U}_1$, $\mathcal{U}_2$, the $\phi_{(a)}$'s from neighboring patches can differ by a transition function $g_{12}(x, y)$:

$$
\begin{aligned}
\phi_{(1)}(t, x, y) &= \phi_{(2)}(t, x, y) + g_{12}(x, y), \\
g_{12}(x, y) &= 2\pi w_{12}^x(x) + 2\pi w_{12}^y(y), \qquad (x, y) \in \mathcal{U}_1 \cap \mathcal{U}_2,
\end{aligned}
\tag{3.12}
$$

where $w_{12}^i(x^i) \in \mathbb{Z}$ are integer-valued, piecewise constant functions defined in $\mathcal{U}_1 \cap \mathcal{U}_2$. On the triple overlaps $\mathcal{U}_1 \cap \mathcal{U}_2 \cap \mathcal{U}_3$ between three patches, the transition functions are subject to the cocycle condition:

$$
g_{12} + g_{23} + g_{31} = 0. \tag{3.13}
$$

The transition function $g_{12}(x, y)$ is generally also not a single-valued function, but is itself a section that requires transition functions. Since the transition function can have its own transition function, the ordinary winding charge $\oint dx^i \partial_i \phi$ is not well-defined.

Let us make this more concrete for a rectangular 2-torus $\mathbb{T}^2$ of lengths $\ell^x, \ell^y$. The $x, y$ coordinates are periodically identified, $x \sim x + \ell^x$, $y \sim y + \ell^y$. We define the integral

$$
c(\phi) \equiv \frac{1}{2\pi} \oint dx \oint dy \, \partial_x \partial_y \phi \tag{3.14}
$$

over the spatial 2-torus $\mathbb{T}^2$. Note that $c(\phi)$ is the total winding dipole charge:

$$
c(\phi) = \oint dx \, Q_x^{xy}(x) = \oint dy \, Q_y^{xy}(y). \tag{3.15}
$$

$c(\phi)$ is analogous to the characteristic class that measures how "twisted" the bundle is. More specifically, $c(\phi)$ measures the winding of the transition function. In particular, if $\phi$ is a globally single-valued (possibly discontinuous) function, or if the transition functions are single-valued, then this integral vanishes.

Let us explore different classes of configurations for $\phi$ in a hierarchical order.

1. $\phi$ is smooth in each patch. At the overlap between two patches, the transition function is therefore a constant, $g_{12}(x, y) = 2\pi n$, $n \in \mathbb{Z}$. Such configurations have finite action and are similar to the configurations in an ordinary theory of compact boson. A general gauge transformation (3.11) would bring such a configuration outside this class.

2. $\phi$ is not necessarily smooth inside a patch, but $\partial_x \partial_y \phi$ is well-defined and finite. For example, $\phi$ depending only on $x$, i.e. $\phi = f(x)$ with periodic $e^{i\phi} = e^{if(x)}$, but with discontinuous $f(x)$. Such configurations do not carry winding dipole charge. Despite the discontinuities, such configurations still have finite action since $\partial_x \partial_y \phi$ is finite.

3. $\phi$ is not necessarily smooth in each patch and $\partial_x \partial_y \phi$ can have a $\delta$-function singularity in $x$ or in $y$, but not in both of them. Now the transition functions can be discontinuous, $g_{12}(x, y) = 2\pi w_{12}^x(x) + 2\pi w_{12}^y(y)$. Furthermore, the transition functions might not be single-valued and require their own transition functions. Such configurations have infinite actions of order $\delta(0) \sim \frac{1}{a}$, but we will argue that they are still meaningful in the continuum field theory.

Let us contrast the above three classes of configurations for the continuum field theory with a typical lattice configuration. On the lattice, $\phi_s$ is subject to the identification $\phi_s \sim \phi_s + 2\pi w(\hat{x}, \hat{y})$ for any $w(\hat{x}, \hat{y}) \in \mathbb{Z}$. A typical lattice configuration has $\Delta_{xy} \phi_s \sim 1$, leading to a divergent action of order $1/a^2$ in the continuum limit. In this limit, we exclude such configurations, but we still study configurations whose $\Delta_{xy} \phi_s \sim a$ or $\Delta_{xy} \phi_s \sim a^2$.

We discuss two examples of configurations for $\phi$ that belong to the third class.

**Example 1** Consider

$$\phi(t,x,y) = 2\pi \frac{x}{\ell^x} W^y(y) + 2\pi \frac{y}{\ell^y} W^x(x) \tag{3.16}$$

where $W^i(x^i) \in \mathbb{Z}$. Across $x = 0$, the transition functions is $g_{(x)}(y) = 2\pi W^y(y)$. Similarly, across $y = 0$, the transition functions is $g_{(y)}(x) = 2\pi W^x(x)$. At the quadruple overlap around $x = y = 0$, since $W^i(x^i)$ are single-valued, discontinuous functions, these transition functions obey the cocycle condition automatically. The integral $c(\phi)$ vanishes for this configuration of $\phi$.

**Example 2** Consider the function on the two-torus

$$\phi(t,x,y) = 2\pi \left[ \frac{x}{\ell^x} \Theta(y - y_0) + \frac{y}{\ell^y} \Theta(x - x_0) - \frac{xy}{\ell^x \ell^y} \right], \tag{3.17}$$

for some $0 < x_0 < \ell^x$ and $0 < y_0 < \ell^y$.[15] We view (3.17) as a function of $-\epsilon \leq x \leq \ell^x + \epsilon$ and $-\epsilon \leq y \leq \ell^y + \epsilon$ for infinitesimal positive $\epsilon$. This function is not single-valued on the torus and needs nontrivial transition functions on the overlaps across both $x = 0$ and $y = 0$

$$\begin{aligned} g_{(x)}(y) &= \phi(x = \ell^x, y) - \phi(x = 0, y) = 2\pi\Theta(y - y_0), \\ g_{(y)}(x) &= \phi(x, y = \ell^y) - \phi(x, y = 0) = 2\pi\Theta(x - x_0). \end{aligned} \tag{3.18}$$

These transition functions are also defined for $-\epsilon \leq x \leq \ell^x + \epsilon$ and $-\epsilon \leq y \leq \ell^y + \epsilon$ and they need their own transition functions, which satisfy a cocycle condition

$$g_{(x)}(y = \ell^y) - g_{(x)}(y = 0) = g_{(y)}(x = \ell^x) - g_{(y)}(x = 0) = 2\pi. \tag{3.19}$$

As advocated previously, the ordinary winding charge, say in the $x$ direction, is $\frac{1}{2\pi} \oint dx \partial_x \phi = \Theta(y - y_0)$, which is not single-valued and ill-defined. We will see that the winding dipole charges are well-defined for this configuration in Section 4.2.

The integral $c(\phi)$ for this configuration is

$$c(\phi) = \oint dx \oint dy \left[ \frac{1}{\ell^x} \delta(y - y_0) + \frac{1}{\ell^y} \delta(x - x_0) - \frac{1}{\ell^x \ell^y} \right] = 1. \tag{3.20}$$

More generally, $c(\phi)$ can be expressed in terms of the transition functions as

$$c(\phi) = \frac{1}{2\pi} \left[ g_{(x)}(y = \ell^y) - g_{(x)}(y = 0) \right] = \frac{1}{2\pi} \left[ g_{(y)}(x = \ell^x) - g_{(y)}(x = 0) \right]. \tag{3.21}$$

# 4 Momentum and Winding Modes

In this section we discuss the spectrum of states of the continuum $\phi$-theory with the minimal Lagrangian in (2.7). In addition to states with energy of order one, we will also discuss states with energy of order $\frac{1}{a}$, which become infinite in the continuum limit. More specifically, these are the lowest energy states carrying a conserved charge.

Following the discussion in Section 1.3, we can further include higher derivative terms respecting the global symmetry of the model. While such higher derivative terms do not affect the generic plane waves, they do change the quantitative behaviors for the charged states. Nonetheless, they will not affect the qualitative features such as the $1/a$ scaling of the energy for these states.

---

[15]When $x_0 = 0$ (and similarly for $y_0 = 0$), more care is needed. Unlike for generic $x_0$, there is now a discontinuity at $x = 0$, even taking into account of transition functions that shift $\phi$ by $2\pi\mathbb{Z}$. This discontinuity leads to $\partial_x \partial_y \phi = 2\pi \left[ \frac{1}{\ell^x} \delta(y - y_0) + \frac{1}{\ell^y} \delta(x) - \frac{1}{\ell^x \ell^y} \right]$, which also gives $c(\phi) = 1$ (3.20).

## 4.1 Momentum Modes

Let us consider a plane wave mode in $\mathbb{R}^{2,1}$:

$$\phi = C e^{i\omega t + i k_x x + i k_y y}. \tag{4.1}$$

The equation of motion (2.8) gives the dispersion relation

$$\omega^2 = \frac{1}{\mu \mu_0} k_x^2 k_y^2. \tag{4.2}$$

For generic $k_x$ and $k_y$ the spectrum is standard, but with a nonstandard dispersion relation. We will discuss it in detail soon.

Classically, the zero-energy solutions $\omega = 0$ are those modes with $k_x = 0$ *or* $k_y = 0$. In particular, there are classical zero-energy solutions with $k_x = 0$ but arbitrarily large $k_y$, and vice versa. The momentum dipole symmetry (2.9) maps one such zero-energy classical solution to another. For this reason we will call these modes the momentum modes. Therefore, classically, the momentum dipole symmetry appears to be spontaneously broken. As we will soon see, this picture is incorrect quantum mechanically.

Note also that the winding symmetry (2.14) vanishes on the plane waves (4.1) and therefore this symmetry does not act on the corresponding states.

Let us quantize the $\phi$ theory on a 2-torus of lengths $\ell^x, \ell^y$, so that the momenta are quantized, i.e. $k_i = 2\pi \frac{n_i}{\ell^i}$ with $n_i \in \mathbb{Z}$. Written in momentum space, the Lagrangian is

$$L = \ell^x \ell^y \sum_{n_x, n_y \in \mathbb{Z}} \left[ \frac{\mu_0}{2} \partial_0 \phi_{n_x, n_y} \partial_0 \phi_{-n_x, -n_y} - \frac{2\pi^2}{\mu} \frac{n_x^2 n_y^2}{(\ell^x \ell^y)^2} \phi_{n_x, n_y} \phi_{-n_x, -n_y} \right], \tag{4.3}$$

where $\phi_{n_x, n_y}$ are the Fourier modes of $\phi$. The quantization of modes $\phi_{n_x, n_y}$ with $n_x \neq 0$ and $n_y \neq 0$ is straightforward. Each such mode behaves as a simple harmonic oscillator with ground state energy

$$E = \frac{\pi}{\sqrt{\mu \mu_0}} \frac{|n_x n_y|}{\ell^x \ell^y}, \quad n_x \neq 0, \, n_y \neq 0. \tag{4.4}$$

On top of this ground state we have a Fock space of states of $\phi$ quanta. Other than the strange dispersion relation, this part of the spectrum is standard.

Let us turn to the modes with $n_x n_y = 0$. As we said above, the momentum symmetry acts on these modes and therefore we refer to them as momentum modes.

For these momentum modes the restoring force of the harmonic oscillator vanishes. Therefore, the corresponding modes can make large field excursions and we need to take into account the identification (3.11). To make the identification manifest, we return to the position space, and focus on the modes with either $n_x = 0$ or $n_y = 0$:

$$\phi(t, x, y) = \phi^x(t, x) + \phi^y(t, y) + \cdots, \tag{4.5}$$

where the $\cdots$ are those modes with $n_x \neq 0$ and $n_y \neq 0$. $\phi^x(t, x), \phi^y(t, y)$ are point-wise $2\pi$ periodic by (3.11):

$$\phi^i(t, x^i) \to \phi^i(t, x^i) + 2\pi w^i(x^i), \quad w^i(x^i) \in \mathbb{Z} \tag{4.6}$$

They share a common zero mode, which implies the following gauge transformation

$$\phi^x(t, x) \to \phi^x(t, x) + c(t), \quad \phi^y(t, y) \to \phi^y(t, y) - c(t). \tag{4.7}$$

The Lagrangian of these momentum modes is

$$L = \frac{\mu_0}{2} \left[ \ell^y \oint dx \, (\dot{\phi}^x)^2 + \ell^x \oint dy \, (\dot{\phi}^y)^2 + 2 \oint dx \, \dot{\phi}^x \oint dy \, \dot{\phi}^y \right] \tag{4.8}$$

It is easy to check that it is consistent with the gauge symmetry (4.7).

The conjugate momenta are

$$\pi^x(t,x) = \mu_0 \left( \ell^y \dot{\phi}^x(t,x) + \oint dy \, \dot{\phi}^y(t,y) \right),$$

$$\pi^y(t,y) = \mu_0 \left( \ell^x \dot{\phi}^y(t,y) + \oint dx \, \dot{\phi}^x(t,x) \right). \tag{4.9}$$

They are subject to the constraint:

$$\oint dx \, \pi^x(x) = \oint dy \, \pi^y(y), \tag{4.10}$$

which can be thought of as Gauss law from the gauge symmetry (4.7). In fact, the momenta are the charges of the momentum dipole symmetry, $Q^i(x^i) = \pi^i(x^i)$.

The point-wise periodicity of $\phi^i$ implies that their conjugate momenta $\pi^i$ are linear combination of delta functions with integer coefficients:

$$Q^x(x) = \pi^x = \sum_\alpha N^x_\alpha \delta(x - x_\alpha), \quad Q^y(y) = \pi^y = \sum_\beta N^y_\beta \delta(y - y_\beta),$$

$$N \equiv \sum_\alpha N^x_\alpha = \sum_\beta N^y_\beta, \quad N^x_\alpha, N^y_\beta \in \mathbb{Z}. \tag{4.11}$$

Here $\{x_\alpha\}$ and $\{y_\beta\}$ are a finite set of points on the $x$ and $y$ axes, respectively.

The Hamiltonian is easily found to be

$$H = \oint dx \, \pi^x \dot{\phi}^x + \oint dy \, \pi^y \dot{\phi}^y - L$$

$$= \frac{1}{2\mu_0 \ell^x \ell^y} \left[ \ell^x \oint dx \, (\pi^x)^2 + \ell^y \oint dy \, (\pi^y)^2 - \left( \oint dx \, \pi^x \right) \left( \oint dy \, \pi^y \right) \right]. \tag{4.12}$$

To check it, substitute (4.9) in (4.12) to express it in terms of $\dot{\phi}$ to find the Lagrangian (4.8).

The configuration with the lowest momentum dipole symmetry charge is

$$\pi^x = \delta(x - x_0), \quad \pi^y = \delta(y - y_0) \tag{4.13}$$

for some $x_0, y_0$. It has energy

$$\frac{1}{2\mu_0 \ell^x \ell^y} \left[ \ell^x \delta(0) + \ell^y \delta(0) - 1 \right]. \tag{4.14}$$

More generally, the energy of the momentum mode (4.11) is

$$H = \frac{1}{2\mu_0 \ell^x \ell^y} \left[ \ell^x \sum_\alpha (N^x_\alpha)^2 \delta(0) + \ell^y \sum_\beta (N^y_\beta)^2 \delta(0) - N^2 \right]. \tag{4.15}$$

We see that in the quantum theory, the energy of the momentum modes is infinite.

To regularize this infinity, we can place the theory on a lattice with spacing $a$.[16] Then the energy of the momentum modes scales as

$$\frac{1}{\mu_0}\frac{1}{\ell a}.\tag{4.16}$$

In the continuum limit $a \to 0$, the momentum modes are much heavier than the generic modes (4.4) whose energies scale as $1/(\mu_0 \ell^2)$. In finite volume, the ground state is the unique eigenstate whose $\pi^i(x^i)$ eigenvalues are all zero, and the classically zero-energy configurations with either $n_x = 0$ or $n_y = 0$ are all lifted quantum mechanically.

In addition, there are momentum modes whose $\pi^i$ includes infinitely many delta functions. For example, if the number of delta functions is of order $1/a$, the energy of such momentum modes scales as $1/a^2$ in the continuum limit. Even though it is of the same order as a typical lattice excitation, the analysis of these modes in the continuum is still meaningful because they carry nontrivial conserved charges.

Let us now consider the infinite volume limit. From the continuum point of view, it is natural to first take $a \to 0$, and then $\ell \to \infty$. In this order of limits, the momentum modes are all lifted, and the energies of the generic modes are brought down to zero as we take $\ell \to \infty$.

We conclude that, in finite volume, the momentum dipole symmetry (2.9), which appears to be spontaneously broken in the classical theory, is in fact restored in the quantum theory. Not only is it restored, but all the states carrying its charge, have infinite energy in the strict continuum limit. Yet, we can still make sense of them as in (4.15).

If instead we first take $\ell \to \infty$ and then $a \to 0$, the momentum modes are still heavier than the generic modes, but their energies both go to zero.

## 4.2  Winding Modes

The most general winding configuration can be obtained by taking linear combinations of (3.17):

$$\phi(t,x,y) = 2\pi \left[ \frac{x}{\ell^x}\left(\sum_\beta W_\beta^y \Theta(y - y_\beta)\right) + \frac{y}{\ell^y}\left(\sum_\alpha W_\alpha^x \Theta(x - x_\alpha)\right) - W\frac{xy}{\ell^x\ell^y}\right],$$
$$W = \sum_\alpha W_\alpha^x = \sum_\beta W_\beta^y, \quad W_\alpha^x, W_\beta^y \in \mathbb{Z},\tag{4.17}$$

where $\{x_\alpha\}$ is a finite set of points between 0 and $\ell^x$, and similarly for the $y_\beta$'s.

This configuration realizes the winding dipole charges:

$$Q_x^{xy}(x) = \frac{1}{2\pi}\oint dy\, \partial_x \partial_y \phi = \sum_\alpha W_\alpha^x\, \delta(x - x_\alpha),$$
$$Q_y^{xy}(y) = \frac{1}{2\pi}\oint dx\, \partial_x \partial_y \phi = \sum_\beta W_\beta^y\, \delta(y - y_\beta).\tag{4.18}$$

The charges are sum of delta functions with integer coefficients. It has a nontrivial $c(\phi)$:

$$c(\phi) = \oint dx \oint dy \left[\frac{1}{\ell^x}\left(\sum_\beta W_\beta^y \delta(y - y_\beta)\right) + \frac{1}{\ell^y}\left(\sum_\alpha W_\alpha^x \delta(x - x_\alpha)\right) - W\frac{1}{\ell^x\ell^y}\right] = W.\tag{4.19}$$

---

[16]Note that the underlying $2 + 1$-dimensional system is in the continuum. Only the $1 + 1$-dimensional system (4.8) is placed on a lattice.

The Hamiltonian of these winding modes is

$$
\begin{aligned}
H &= \frac{1}{2\mu} \oint dx \oint dy (\partial_x \partial_y \phi)^2 \\
&= \frac{2\pi^2}{\mu} \left[ \frac{1}{\ell^y} \oint dx (W_\alpha^x)^2 \delta(x-x_\alpha)^2 + \frac{1}{\ell^x} \oint dy (W_\beta^y)^2 \delta(y-y_\beta)^2 - \frac{W^2}{\ell^x \ell^y} \right] \\
&= \frac{2\pi^2}{\mu \ell^x \ell^y} \left[ \ell^x \sum_\alpha (W_\alpha^x)^2 \delta(0) + \ell^y \sum_\beta (W_\beta^y)^2 \delta(0) - W^2 \right] .
\end{aligned}
\tag{4.20}
$$

The energy of the winding mode is infinite in the continuum limit. To regularize this infinity, we can place the theory on a lattice with spacing $a$, then the energy of the winding state scales as $1/(\mu \ell a)$.

There are also winding modes whose charges $Q_i^{xy}$ involve infinitely many delta functions. Such winding modes have energy of order $1/a^2$.

## 5 Self-Duality

Let us rewrite the Euclidean Lagrangian for $\phi$ as

$$
\mathcal{L}_E = \frac{\mu_0}{2} B^2 + \frac{1}{2\mu} E_{xy} E^{xy} + \frac{i}{2\pi} \widetilde{B}^{xy} (\partial_x \partial_y \phi - E_{xy}) + \frac{i}{2\pi} \widetilde{E} (\partial_\tau \phi - B)
\tag{5.1}
$$

where $B, E_{xy}, \widetilde{E}, \widetilde{B}^{xy}$ are independent fields. We denote the Euclidean time as $\tau$. If we integrate out these fields, we recover the original Lagrangian (2.7) for $\phi$.

Instead, we integrate out only $B, E_{xy}$

$$
\mathcal{L}_E = \frac{1}{8\pi^2 \mu_0} \widetilde{E}^2 + \frac{\mu}{8\pi^2} \widetilde{B}_{xy} \widetilde{B}^{xy} + \frac{i}{2\pi} \widetilde{B}^{xy} \partial_x \partial_y \phi + \frac{i}{2\pi} \widetilde{E} \partial_\tau \phi
\tag{5.2}
$$

Next, we integrate out $\phi$ to find the constraint

$$
\partial_\tau \widetilde{E} = \partial_x \partial_y \widetilde{B}^{xy} .
\tag{5.3}
$$

This can be solved locally in terms of a field $\phi^{xy}$ with spin 2 under the spatial $\mathbb{Z}_4$:

$$
\begin{aligned}
\widetilde{E} &= \partial_x \partial_y \phi^{xy} , \\
\widetilde{B}^{xy} &= \partial_\tau \phi^{xy} .
\end{aligned}
\tag{5.4}
$$

The Lagrangian becomes

$$
\mathcal{L}_E = \frac{\widetilde{\mu}_0}{2} (\partial_\tau \phi^{xy})^2 + \frac{1}{2\widetilde{\mu}} (\partial_x \partial_y \phi^{xy})^2 ,
\tag{5.5}
$$

where

$$
\widetilde{\mu}_0 = \frac{\mu}{4\pi^2} , \quad \widetilde{\mu} = 4\pi^2 \mu_0 .
\tag{5.6}
$$

Hence the $\phi$ theory is dual to a theory of $\phi^{xy}$, which transforms in the spin 2 representation of the spatial $\mathbb{Z}_4$ symmetry.

Let us clarify why we refer to this as self-duality. Since the spatial rotation symmetry is discrete, it can be redefined by discrete internal global symmetries. Both the $\phi$ and the $\phi^{xy}$ theories have a charge conjugation symmetry, $C: \phi \to -\phi$, $C: \phi^{xy} \to -\phi^{xy}$. The unitary

global symmetry is therefore $\mathbb{Z}_2^C \times \mathbb{Z}_4$. Let $R$ be the generator of the spatial $\mathbb{Z}_4$ rotation, $R^4 = 1$. $\phi$ has spin 0 under $R$ and $\phi^{xy}$ has spin 2 under $R$. However, if we say that the rotation generator of the $\phi^{xy}$ theory is $\widetilde{R} \equiv RC$, then $\phi^{xy}$ has spin 0 under $\widetilde{R}$, i.e., the sels-duality maps $R \longleftrightarrow RC$. More abstractly, the representations for $\phi$ and $\phi^{xy}$ are related by an outer automorphism of the $\mathbb{Z}_2^C \times \mathbb{Z}_4$ symmetry group. Similar nontrivial maps of representations by outer automorphisms are common in dualities.

$\phi^{xy}$ is subject to the identification

$$\phi^{xy}(t,x,y) \sim \phi^{xy}(t,x,y) + 2\pi w^x(x) + 2\pi w^y(y), \tag{5.7}$$

where $w^i(x^i) \in \mathbb{Z}$. Similarly, the local operators include $\partial_0 \phi^{xy}, \partial_x \partial_y \phi^{xy}, e^{i\phi^{xy}}$, but not $\partial^i \phi^{xy}$. Just like the $\phi$ field, $\phi^{xy}$ is generally a section over a bundle with nontrivial transition functions.

There is no ordinary winding symmetry of $\phi^{xy}$ because $\partial^i \phi^{xy}$ is not a well-defined operator. Even though there is no ordinary winding symmetry, there is a dual winding dipole symmetry:

$$J_0 = \frac{1}{2\pi}\partial_x \partial_y \phi^{xy}, \qquad J^{xy} = \frac{1}{2\pi}\partial_0 \phi^{xy},$$
$$\partial_0 J_0 = \partial_x \partial_y J^{xy}, \tag{5.8}$$

with $(\mathbf{R}_{\text{time}}, \mathbf{R}_{\text{space}}) = (\mathbf{1}_0, \mathbf{1}_2)$. This is dual to the momentum dipole symmetry (2.9) of $\phi$.

The equation of motion in the Lorentzian signature

$$\widetilde{\mu}_0 \partial_0^2 \phi^{xy} = -\frac{1}{\widetilde{\mu}}\partial_x^2 \partial_y^2 \phi^{xy}, \tag{5.9}$$

implies a dual momentum dipole symmetry

$$J_0^{xy} = \widetilde{\mu}_0 \partial_0 \phi^{xy}, \qquad J = -\frac{1}{\widetilde{\mu}}\partial_x \partial_y \phi^{xy},$$
$$\partial_0 J_0^{xy} = \partial^x \partial^y J, \tag{5.10}$$

with $(\mathbf{R}_{\text{time}}, \mathbf{R}_{\text{space}}) = (\mathbf{1}_2, \mathbf{1}_0)$. This is dual to the winding dipole symmetry (2.13) of $\phi$.

Under the duality, the momentum modes (4.11) of $\phi$ are mapped to the winding modes (4.17) of $\phi^{xy}$ by $N_\alpha^i \longleftrightarrow W_\alpha^i$. The momentum dipole charges and the winding dipole charges are exchanged $Q^i \longleftrightarrow Q_i^{xy}$. Indeed, the Hamiltonian of the momentum modes (4.15) agrees with that of the winding (4.20) under the above mapping and (5.6).

Our self-duality, which is wrong on the lattice, is true not only in the strict continuum limit, but also for the charged states with energy of order $\frac{1}{a}$.

# 6 Robustness and Universality

Since we consider discontinuous field configurations, higher derivative terms might not be suppressed and the universality of the leading order terms in our Lagrangian (2.7) could be affected. Let us discuss it in more detail.

We start with the momentum modes in Section 4.1. In addition to the leading order terms in (2.7), we can add to the Lagrangian higher derivative terms. In the spirit of naturalness, we limit ourselves to terms that preserve the momentum dipole global symmetry, e.g.,

$$g(\partial_0 \partial_x \phi)^2. \tag{6.1}$$

Since this term has more derivatives than the leading order term, its coefficient $g$ will be taken to be of order $a^2$. Therefore, it has a negligible effect on any generic plane wave mode of finite

energy in the $a \to 0$ limit. This is not true for the momentum modes, where $\pi \sim \partial_0 \phi$ is a sum of delta functions. Therefore, the contribution from the higher derivative term (6.1) to the momentum modes is not suppressed by the additional power of derivative $\partial_x$. More precisely, such a higher derivative term changes the energy of a momentum mode by order $g/a^3 \sim 1/a$.

We conclude that while the precise energy of these momentum modes is subject to corrections from the higher derivative terms, their scaling in $1/a$ is universal. A similar conclusion holds for the other heavier momentum modes involving infinitely many delta functions.

This example demonstrates that in this case the expansion in the power of derivatives might not be valid. Logically, there could be a precise cancellation between different terms each contributing at order $1/a$. However, such a cancellation is not natural and depends on fine tuning of the parameters.

Next, we discuss the effects of higher derivative terms on the winding modes in Section 4.2. To be concrete, let us consider adding

$$g(\partial_x^2 \partial_y \phi)^2 \tag{6.2}$$

to the minimal Lagrangian (2.7), with the coupling $g$ of order $a^2$.[17] As in the discussion around (6.1), this term respects all the symmetries of the problem and it has negligible effect on the generic plane wave modes of finite energy. The winding modes, by contrast, have discontinuous $\phi$ and delta functions in $\partial_x \partial_y \phi$, and therefore the correction to their energy is not suppressed by the additional derivative $\partial_x$. More specifically, the contribution to the energy of a typical winding mode from this term is of order $g/a^3 \sim 1/a$. Therefore, as in the discussion of the momentum modes, the precise, quantitative results for the energy of the winding modes are not universal, but their qualitative scaling in $1/a$ is. A similar conclusion holds for the other heavier winding modes involving infinitely many delta functions.

Finally, we discuss the robustness of the $\phi$-theory (2.7). In the XY-plaquette lattice model, we impose the $(\mathbf{1}_0, \mathbf{1}_2)$ momentum dipole symmetry as our microscopic symmetry $G_{UV}$. The continuum field theory of $\phi$ has a larger global symmetry $G_{IR}$ that includes not only $G_{UV}$, but also the $(\mathbf{1}_2, \mathbf{1}_0)$ winding dipole symmetry. The $G_{UV}$-invariant operators include $e^{i\phi^{xy}}$, which violates the emergent $(\mathbf{1}_2, \mathbf{1}_0)$ winding dipole symmetry. Such operators can affect the robustness of the theory. However, as discussed in Section 4.2, the state created by $e^{i\phi^{xy}}$ has energy of order $\frac{1}{a}$. Therefore, $e^{i\phi^{xy}}$ is a trivial operator in the low-energy limit. It is very irrelevant and cannot affect the robustness. This is to be contrasted with the ordinary $1+1$-dimensional compact boson at small radius where the relevant winding operator destabilizes the conformal field theory.

In fact, we will now argue that our low energy theory is even more robust. It is robust even under deformations that violate the $(\mathbf{1}_0, \mathbf{1}_2)$ momentum dipole symmetry.

The simplest operators violating this symmetry are of the form $e^{i\phi}$. They create momentum modes with energy of order $1/a$. Therefore, these operators are very irrelevant. (See Appendix A, for a computation of the two-point functions of $e^{i\phi}$, which demonstrates it.) Therefore, deforming the low-energy theory by operators like $e^{i\phi}$ does not affect the long distance behavior.

One might question the robustness of the theory under deformations by operators of the form $\partial_x \phi$, or by rotation invariant operators like $(\partial_x \phi)^2 + (\partial_y \phi)^2$. From the low-energy point of view, these operators are not well-defined, because they are not invariant under the gauge transformation (3.11). Therefore, they are not allowed deformations.

However, one might still question the robustness under deformations of the underlying lattice model by operators of the form

$$e^{i\phi_{\hat{x}+1,\hat{y}}} e^{-i\phi_{\hat{x},\hat{y}}} , \tag{6.3}$$

---

[17]In fact, under the duality in Section 5, this higher derivative term is dual to the term $g(\partial_0 \partial_x \phi)^2$ (6.1).

i.e., standard nearest neighbor coupling of the microscopic spins. This operator violates the global symmetry, but it is a well-defined, gauge-invariant operator. In the continuum limit of the standard XY model, this operator becomes $e^{i\phi(x+a,y)}e^{-i\phi(x,y)} = 1 + ia\partial_x\phi(x,y) + i\frac{a^2}{2}\partial_x^2\phi(x,y) - \frac{a^2}{2}(\partial_x\phi(x,y))^2 + \mathcal{O}(a^3)$ and after renormalization flows to operators of the form $\partial_x^k\phi$. In our case, $\partial_x^k\phi$ are not a valid operators, but the continuum limit of (6.3) should still make sense. It violates the momentum symmetry and can ruin the low-energy theory.

We claim that the continuum limit of (6.3) is also very irrelevant. One way to understand it is to note that the product of operators

$$e^{in\phi(x,y)}e^{in'\phi(x',y')} \tag{6.4}$$

does not have a standard operator product expansion starting with $e^{i(n+n')\phi(x,y)}$. The product (6.4) carries different subsystem symmetry charges than $e^{i(n+n')\phi(x,y)}$. Therefore, we cannot define operators like $\partial_x\phi$ using this separated points product. Related to that, the product (6.4) creates states carrying the subsystem symmetry. Such states have energy of order $1/a$ and therefore they are also very irrelevant. In Appendix A, we will study the two-point functions of (6.4) and will check this assertion in more detail.

We conclude that we can start at short distances with an arbitrary theory without the momentum symmetry. Then, we can fine tune the parameters to find at low energies the $\phi$-theory (2.7). Once we find this low-energy theory, small deformations of the UV theory translate to small deformations of the IR theory. Since all these operators are irrelevant, the IR theory is robust!

# 7 $U(1)$ Tensor Gauge Theory

In this section we study a tensor gauge theory in $2+1$ dimensions. Its gauge symmetry is a local version of the global $U(1)$ dipole symmetry we discussed above. As we will see, it exhibits peculiarities that are not present in ordinary $U(1)$ gauge theories. We will also see that in many ways it is reminiscent of an ordinary $U(1)$ gauge theory in $1+1$ dimensions. In [1] we will study a similar gauge theory in $3+1$ dimensions.

We can gauge the $(\mathbf{1}_0, \mathbf{1}_2)$ momentum dipole global symmetry of the $\phi$-theory by coupling the currents to the tensor gauge field $(A_0, A_{xy})$:

$$J_0 A_0 + J^{xy} A_{xy}. \tag{7.1}$$

The current conservation equation $\partial_0 J_0 = \partial_x \partial_y J^{xy}$ implies the gauge transformation

$$
\begin{aligned}
A_0 &\to A_0 + \partial_0 \alpha, \\
A_{xy} &\to A_{xy} + \partial_x \partial_y \alpha.
\end{aligned}
\tag{7.2}
$$

The gauge invariant electric field is

$$E_{xy} = \partial_0 A_{xy} - \partial_x \partial_y A_0, \tag{7.3}$$

while there is no magnetic field.

## 7.1 Lattice Tensor Gauge Theory

Let us discuss the lattice version of the $U(1)$ tensor gauge theory without matter. We have a $U(1)$ phase variable $U_p = e^{ia^2 A_p}$ and its conjugate variable $E_p$ at every plaquette. The gauge

transformation $e^{i\alpha_s}$ is a $U(1)$ phase associated with each site $s$. Under the gauge transformation,

$$U_p \to U_p \, e^{i\Delta_{xy}\alpha_s} \tag{7.4}$$

where $\Delta_{xy}\alpha_s$ is a linear combination of $\alpha_s$ around the plaquette $p$.

There are two types of gauge invariant operators. The first type is an operator $E_p$ at a single plaquette. The second type is a product of $U_p$'s along the $x$ direction at a fixed $y$, or vice versa.

Gauss law sets

$$G_s \equiv \sum_{p \ni s} \epsilon_p E_p = 0 \tag{7.5}$$

where the sum is an oriented sum ($\epsilon_p = \pm 1$) over the four plaquettes $p$ that share a common site $s$. The Hamiltonian is

$$H = \frac{1}{g^2} \sum_p E_p^2, \tag{7.6}$$

with Gauss law imposed by hand.

The lattice model has an electric tensor symmetry whose conserved charge is proportional to $E_p$. Clearly it commutes with the Hamiltonian, which depends only on $E_p$. The electric tensor symmetry rotates the phase of $U_p$ at a single plaquette, $U_p \to e^{i\varphi} U_p$. Using Gauss law (7.5), the dependence of the conserved charge $Q_p$ on $p$ is a function of $\hat{x}$ plus a function of $\hat{y}$.

## 7.2 Lagrangian

Motivated by earlier papers about related models, this gauge theory was studied in [22, 23, 31, 33]. The Lorentzian Lagrangian of the pure tensor gauge theory is

$$\mathcal{L} = \frac{1}{g_e^2} E_{xy}^2 + \frac{\theta}{2\pi} E_{xy}. \tag{7.7}$$

Note that $g_e$ has mass dimension $\frac{3}{2}$ and $\theta$ is dimensionless.

We will soon show that the total electric flux in Euclidean space is quantized $\oint d\tau dx dy \, E_{xy} \in 2\pi\mathbb{Z}$, and therefore the theta angle is $2\pi$ periodic $\theta \sim \theta + 2\pi$.

The equations of motion are

$$\begin{aligned} \partial_0 E_{xy} &= 0, \\ \partial^x \partial^y E_{xy} &= 0, \end{aligned} \tag{7.8}$$

where the second equation is Gauss law.

## 7.3 Fluxes

We place the theory on a Euclidean 3-torus with lengths $\ell^x, \ell^y, \ell^\tau$ and explore its bundles. For that, we need to understand the possible nontrivial transition functions.

Recalling the winding configuration (3.17), we take the transition function at $\tau = \ell^\tau$ to be a gauge transformation with

$$g(x, y) = 2\pi \left[ \frac{x}{\ell^x}\Theta(y - y_0) + \frac{y}{\ell^y}\Theta(x - x_0) - \frac{xy}{\ell^x \ell^y} \right], \tag{7.9}$$

i.e.

$$A_{xy}(\tau = \ell^\tau, x, y) = A_{xy}(\tau = 0, x, y) + \partial_x \partial_y g. \tag{7.10}$$

For example, we can have

$$A_{xy}(\tau, x, y) = 2\pi \frac{\tau}{\ell^\tau} \left[ \frac{1}{\ell^x}\delta(y - y_0) + \frac{1}{\ell^y}\delta(x - x_0) - \frac{1}{\ell^x \ell^y} \right], \tag{7.11}$$

Such a configuration gives rise to electric flux:

$$e_{(x)}(x) \equiv \oint d\tau \oint dy\, E_{xy} = 2\pi \delta(x - x_0),$$

$$e_{(y)}(y) \equiv \oint d\tau \oint dx\, E_{xy} = 2\pi \delta(y - y_0). \tag{7.12}$$

With more general such twists we have

$$e_{(x)}(x) = \oint d\tau \oint dy\, E_{xy} = 2\pi \sum_{\alpha} n_{x\alpha} \delta(x - x_\alpha),$$

$$e_{(y)}(y) = \oint d\tau \oint dx\, E_{xy} = 2\pi \sum_{\beta} n_{y\beta} \delta(y - y_\beta), \tag{7.13}$$

$$\sum_{\alpha} n_{x\alpha} = \sum_{\beta} n_{y\beta}, \quad n_{x\alpha}, n_{y\beta} \in \mathbb{Z}$$

or in its integrated form

$$e_{(x)}(x_1, x_2) \equiv \oint d\tau \int_{x_1}^{x_2} dx \oint dy\, E_{xy} \in 2\pi\mathbb{Z},$$

$$e_{(y)}(y_1, y_2) \equiv \oint d\tau \oint dx \int_{y_1}^{y_2} dy\, E_{xy} \in 2\pi\mathbb{Z}. \tag{7.14}$$

These quantized fluxes and their associated transition functions have been previously discussed in [23].

## 7.4 Global Symmetry

The equations of motion can be interpreted as the current conservation equation and a differential condition for an *electric tensor symmetry*:

$$\partial_0 J_0^{xy} = 0,$$
$$\partial_x \partial_y J_0^{xy} = 0, \tag{7.15}$$

with the current in the spin 2 representation $\mathbf{1}_2$ of the spatial $\mathbb{Z}_4$ group:

$$J_0^{xy} = \frac{2}{g_e^2} E_{xy} + \frac{\theta}{2\pi}. \tag{7.16}$$

We define the current with a shift by $\theta/2\pi$ so that the conserved charge is properly quantized (see (7.32)). Note that there is no spatial component of the current. This is analogous to the electric one-form symmetry of the ordinary $1 + 1$-dimensional $U(1)$ gauge theory whose current is $J_0^x = \frac{2}{g^2} E_x + \frac{\theta}{2\pi}$ obeying $\partial_0 J_0^x = 0$ and $\partial_x J_0^x = 0$.

There is an integer conserved charge at every point in space, which coincides with the current itself:

$$Q(x, y) = J_0^{xy} = N^x(x) + N^y(y), \tag{7.17}$$

where $N^i(x^i) \in \mathbb{Z}$. The differential condition $\partial_x \partial_y J_0^{xy} = 0$ constrains the charge $Q$ to be an integer function of $x$ plus an integer function of $y$.

Up to a gauge transformation, the electric tensor symmetry acts on the gauge fields as

$$A_{xy} \to A_{xy} + c^x(x) + c^y(y). \tag{7.18}$$

As a symmetry, it maps one configuration of $A_{xy}$ to another with the same electric field.

This conserved charge exists also on the lattice.

The charged objects under this electric global symmetry are the gauge-invariant extended operators defined at a fixed time:

$$
\begin{aligned}
W_{(x)}(x_1, x_2) &= \exp\left[ i \int_{x_1}^{x_2} dx \oint dy A_{xy} \right], \\
W_{(y)}(y_1, y_2) &= \exp\left[ i \oint dx \int_{y_1}^{y_2} dy A_{xy} \right].
\end{aligned}
\tag{7.19}
$$

Only integer powers of this operator are invariant under the large gauge transformation of the form (7.9). We can refer to such operators as Wilson strips. Note that gauge invariance restricts the allowed positions of the strips. The symmetry operator $\mathcal{U}(\beta; x, y) = e^{i\beta Q(x,y)}$ obeys the following commutation relation with the Wilson strip

$$
\begin{aligned}
\mathcal{U}(\beta; x, y) W_{(x)}(x_1, x_2) &= e^{i\beta} W_{(x)}(x_1, x_2) \mathcal{U}(\beta; x, y), && \text{if } x_1 < x < x_2, \\
\mathcal{U}(\beta; x, y) W_{(y)}(y_1, y_2) &= e^{i\beta} W_{(y)}(y_1, y_2) \mathcal{U}(\beta; x, y), && \text{if } y_1 < y < y_2.
\end{aligned}
\tag{7.20}
$$

These strip operators are the continuum version of the lattice operators constructed as products of $U_p$ along a line.

## 7.5 Defects as Fractons

We now discuss defects that are extended in the time direction. The simplest kind of such a defect is

$$
\exp\left[ i \int_{-\infty}^{\infty} dt A_0 \right].
\tag{7.21}
$$

In Euclidean signature with compact time direction, the exponent is quantized by a gauge transformation $\alpha = 2\pi \frac{\tau}{\ell^\tau}$ that winds nontrivially in the time direction. This describes a single static charged particle. Importantly, a single particle cannot move in space by itself. Gauge invariance makes it immobile.

While a single particle cannot move in isolation, a pair of them with opposite charges – a dipole – can move collectively. Consider two particles with charges $\pm 1$ at fixed $x_1$ and $x_2$ moving in time along a curve $\mathcal{C}$ in the $(y, t)$ plane, $y(t)$. This motion is described by the gauge-invariant defect

$$
W(x_1, x_2, \mathcal{C}) = \exp\left[ i \int_{x_1}^{x_2} dx \int_{\mathcal{C}} \left( dt \partial_x A_0 + dy A_{xy} \right) \right]
\tag{7.22}
$$

Note that the integrand $\int_{\mathcal{C}} \left( dt \partial_x A_0 + dy A_{xy} \right)$ is gauge-invariant for any curve $\mathcal{C}$ without endpoints, e.g. running from the far past to the far future. Similarly, we can have a pair of particles separated in the $y$ directions moving collectively in the $x$ direction.

Finally, the operators (7.19) are special cases of these defects where $\mathcal{C}$ is a closed curve independent of time.

The restricted mobility of these probe particles is the hallmark of fractons.

## 7.6 An Effective Theory and the Spectrum

We place the system on a spatial 2-torus with lengths $\ell^x, \ell^y$ and study its spectrum.

We pick the temporal gauge $A_0 = 0$ and then Gauss law tells us that

$$
\partial^x \partial^y E_{xy} = 0.
\tag{7.23}
$$

It is solved, up to a time independent gauge transformation, by

$$A_{xy} = \frac{1}{\ell^y} f^x(t,x) + \frac{1}{\ell^x} f^y(t,y), \tag{7.24}$$

where the normalization was picked for later convenience. Note that there is no mode with nontrivial momenta in both the $x$ and $y$ directions. This is analogous to the ordinary $1+1$-dimensional $U(1)$ gauge theory where there is no propagating degrees of freedom.

Only the sum of the zero modes of $\frac{1}{\ell^y} f^x(x)$ and $\frac{1}{\ell^x} f^y(y)$ is physical. This implies a gauge symmetry:

$$
\begin{aligned}
f^x(t,x) &\to f^x(t,x) + \ell^y \, c(t), \\
f^y(t,y) &\to f^y(t,y) - \ell^x \, c(t).
\end{aligned} \tag{7.25}
$$

To remove this gauge ambiguity, we define the gauge-invariant variables $\bar f^i$ as

$$
\begin{aligned}
\bar f^x(t,x) &= f^x(t,x) + \frac{1}{\ell^x} \oint dy f^y(t,y), \\
\bar f^y(t,y) &= f^y(t,y) + \frac{1}{\ell^y} \oint dx f^x(t,x).
\end{aligned} \tag{7.26}
$$

The price we pay is that these variables are subject to a constraint

$$\oint dx \bar f^x(t,x) = \oint dy \bar f^y(t,y). \tag{7.27}$$

By performing a gauge transformation $\alpha$ of the form (7.9), we obtain the following two identifications on $\bar f^i$:

$$
\begin{aligned}
\bar f^x(t,x) &\to \bar f^x(t,x) + 2\pi\delta(x - x_0), \\
\bar f^y(t,y) &\to \bar f^y(t,y) + 2\pi\delta(y),
\end{aligned} \tag{7.28}
$$

for each $x_0$, and

$$
\begin{aligned}
\bar f^x(t,x) &\to \bar f^x(t,x), \\
\bar f^y(t,y) &\to \bar f^y(t,y) + 2\pi\delta(y - y_0) - 2\pi\delta(y),
\end{aligned} \tag{7.29}
$$

for each $y_0$. On a lattice with $L^i$ sites in the $x^i$ direction, we can solve the first $\bar f^y(\hat y = 1)$ in terms of the other coordinates using (7.27), then the remaining $L^x + L^y - 1$ $\bar f$'s have periodicities $\bar f \sim \bar f + \frac{2\pi}{a}$.

The Lagrangian for these modes is

$$L = \frac{1}{g_e^2 \ell^x \ell^y} \left[ \ell^x \oint dx (\dot{\bar f}^x)^2 + \ell^y \oint dy (\dot{\bar f}^y)^2 - \left( \oint dx \dot{\bar f}^x \right) \left( \oint dy \dot{\bar f}^y \right) \right] + \frac{\theta}{2\pi} \oint dx \dot{\bar f}^x. \tag{7.30}$$

Let $\bar\Pi^x(x)$ and $\bar\Pi^y(y)$ be the conjugate momenta of $\bar f^i$. The delta function periodicities (7.28) and (7.29) imply that $\bar\Pi^i(x^i)$ have independent integer eigenvalues at every $x^i$. Due to the constraint (7.27) on $\bar f^i$, the conjugate momenta $\bar\Pi^i$ are subject to a gauge ambiguity generated by the constraint:

$$
\begin{aligned}
\bar\Pi^x(x) &\sim \bar\Pi^x(x) + 1, \\
\bar\Pi^y(y) &\sim \bar\Pi^y(y) - 1.
\end{aligned} \tag{7.31}
$$

The charge of the electric global symmetry (7.17) is expressed in terms of the conjugate momenta as

$$Q(x,y) = \frac{2}{g_e^2} E_{xy} + \frac{\theta}{2\pi} = \bar\Pi^x(x) + \bar\Pi^y(y). \tag{7.32}$$

The Hamiltonian is

$$H = \frac{g_e^2}{4}\left[\ell^y \oint dx\left(\bar{\Pi}^x - \frac{\theta_x}{2\pi}\right)^2 + \ell^x \oint dy\left(\bar{\Pi}^y - \frac{\theta_y}{2\pi}\right)^2 \right.$$
$$\left. +2\oint dx\left(\bar{\Pi}^x - \frac{\theta_x}{2\pi}\right)\oint dy\left(\bar{\Pi}^y - \frac{\theta_y}{2\pi}\right)\right], \tag{7.33}$$

where $\theta_x + \theta_y = \theta$. One can show that the Hamiltonian only depends on the sum of $\theta_x, \theta_y$, but not the difference.

Let us regularize this Hamiltonian on a lattice with $L^x, L^y$ sites in the $x, y$ directions, respectively. We will label the lattice site as $(\hat{x}, \hat{y})$ with $\hat{x} = 1, \cdots, L^x$ and $\hat{y} = 1, \cdots, L^y$ and let $a$ be the lattice spacing. The conjugate momenta $\bar{\Pi}^i(\hat{x}^i)$ have independent integer eigenvalues at each site $\hat{x}^i$. The Hamiltonian is

$$H = \frac{g_e^2 a}{4}\left[\ell^y \sum_{\hat{x}=1}^{L^x}\left(\bar{\Pi}^x(\hat{x}) - \frac{\theta_x}{2\pi}\right)^2 + \ell^x \sum_{\hat{y}=1}^{L^y}\left(\bar{\Pi}^y(\hat{y}) - \frac{\theta_y}{2\pi}\right)^2 \right.$$
$$\left. +2a \sum_{\hat{x}=1}^{L^x}\left(\bar{\Pi}^x(\hat{x}) - \frac{\theta_x}{2\pi}\right)\sum_{\hat{y}=1}^{L^y}\left(\bar{\Pi}^y(\hat{y}) - \frac{\theta_y}{2\pi}\right)\right]. \tag{7.34}$$

States with finitely many nonzero $\bar{\Pi}^i(\hat{x})$ have very small energies of order $a$, which vanish in the continuum limit. This is to be contrasted with the $\phi$ theory where the classically zero-energy modes are lifted quantum mechanically. We also have states with order $L$ nonzero $\bar{\Pi}^i(\hat{x})$. For example, in the continuum notation, $\bar{\Pi}^x(x) = \Theta(x - x_1) - \Theta(x - x_2)$ with $x_1 < x_2$. The energies of such states are of order 1.

## 7.7 Robustness and Universality

As in the $\phi$-theory, we now discuss the effects of higher derivative terms on the states in the gauge theory. For example, consider

$$g(\partial_x E_{xy})^2, \tag{7.35}$$

with the coefficient $g$ taken to be of order $a^2$. As we discussed, states with finitely many nonzero $\bar{\Pi} \sim E_{xy}$ have energy of order $a$, which goes to zero in the continuum limit. The term (7.35) shifts their energy by an amount of order $g/a \sim a$. Therefore, the energy of these states remains zero in the continuum limit. States with order $1/a$ nonzero $\bar{\Pi}$ have energy of order one and they receive corrections of order one from terms like (7.35). Therefore, the computation of their energy using the original Lagrangian (7.7) is not universal. To conclude, while the zero-energy states are not lifted by these higher derivative terms, the finite energy states do receive quantitative corrections. Nonetheless, the qualitative features of these charged modes are universal.

Let us discuss the robustness the global symmetry. On the lattice, there is an electric tensor global symmetry $G_{UV}$ (7.15), which coincides with the symmetry $G_{IR}$ of the low-energy field theory. Similar to the ordinary $1 + 1$-dimensional $U(1)$ gauge theory discussed in Section 1.2, there is no relevant operator violating this symmetry. The effect of adding massive charged particles at short distances is similarly negligible in the continuum limit. We conclude that the electric tensor symmetry $G_{UV}$ is robust in the $2 + 1$-dimensional $U(1)$ tensor gauge theory.

# 8   $\mathbb{Z}_N$ Tensor Gauge Theory

In this section we discuss a $\mathbb{Z}_N$ version of the tensor gauge theory of Section 7. The lattice version of this theory is simply a $\mathbb{Z}_N$ version of the lattice model of Section 7. A continuum version of this theory can be obtained by coupling the $U(1)$ theory to a scalar field $\phi$ with charge $N$ that Higgses it to $\mathbb{Z}_N$. This $2+1$-dimensional $\mathbb{Z}_N$ tensor gauge theory is in many ways analogous to the $1+1$-dimensional $\mathbb{Z}_N$ gauge theory.

## 8.1   Lagrangian

The Euclidean Lagrangian is:

$$\mathcal{L}_E = \frac{i}{2\pi}\hat{E}^{xy}(\partial_x\partial_y\phi - NA_{xy}) + \frac{i}{2\pi}\hat{B}(\partial_\tau\phi - NA_\tau), \tag{8.1}$$

where $(A_\tau, A_{xy})$ are the $U(1)$ tensor gauge fields and $\phi$ is a $2\pi$-periodic real scalar field that Higgses the $U(1)$ gauge symmetry to $\mathbb{Z}_N$. The gauge transformations are

$$\begin{aligned}
\phi &\sim \phi + N\alpha, \\
A_\tau &\sim A_\tau + \partial_\tau\alpha, \\
A_{xy} &\sim A_{xy} + \partial_x\partial_y\alpha.
\end{aligned} \tag{8.2}$$

The fields $\hat{E}^{xy}$ and $\hat{B}$ are Lagrangian multipliers. The equations of motion are

$$\begin{aligned}
\partial_x\partial_y\phi - NA_{xy} &= 0, \\
\partial_\tau\phi - NA_\tau &= 0, \\
\hat{E}^{xy} = \hat{B} &= 0.
\end{aligned} \tag{8.3}$$

We can dualize (8.1) by integrating out $\phi$. This leads to the constraint

$$\partial_x\partial_y\hat{E}^{xy} - \partial_\tau\hat{B} = 0, \tag{8.4}$$

which is solved locally in terms of a spin-two field $\phi^{xy}$

$$\hat{E}^{xy} = \partial_\tau\phi^{xy}, \qquad \hat{B} = \partial_x\partial_y\phi^{xy}. \tag{8.5}$$

The winding modes of $\phi$ mean that the periods of $\hat{E}^{xy}$ and of $\hat{B}$ are quantized, corresponding to $\phi^{xy} \sim \phi^{xy} + 2\pi$. Then, (8.1) becomes

$$\mathcal{L}_E = \frac{i}{2\pi}N\phi^{xy}(\partial_\tau A_{xy} - \partial_x\partial_y A_\tau) = \frac{i}{2\pi}N\phi^{xy}E_{xy}. \tag{8.6}$$

The Lagrangian is analogous to the $BF$-type Lagrangian of the $1+1$-dimensional $\mathbb{Z}_N$ gauge theory (1.5). The equations of motion are

$$\begin{aligned}
\partial_\tau\phi^{xy} &= 0, \\
\partial_x\partial_y\phi^{xy} &= 0, \\
E_{xy} &= 0.
\end{aligned} \tag{8.7}$$

## 8.2 Global Symmetry

Let us track the global symmetries of the system. The scalar field theory $\phi$ has a global $U(1)$ momentum dipole symmetry (2.9) and a global $U(1)$ winding dipole symmetry (2.13). The momentum symmetry is gauged and the gauging turns the $U(1)$ winding dipole symmetry into $\mathbb{Z}_N$. Under the duality, the $\mathbb{Z}_N$ winding dipole symmetry of $\phi$ becomes the $\mathbb{Z}_N$ momentum dipole symmetry of $\phi^{xy}$. In addition, the pure gauge theory has an $U(1)$ electric global symmetry (7.15) and the coupling to the matter field $\phi$ breaks it to $\mathbb{Z}_N$. Altogether, we have a $\mathbb{Z}_N$ dipole global symmetry and a $\mathbb{Z}_N$ electric global symmetry.

Let us discuss the gauge-invariant operators at a fixed time. The gauge-invariant local operator

$$e^{i\phi^{xy}} \tag{8.8}$$

is the symmetry operator that generates the $\mathbb{Z}_N$ electric global symmetry. In addition, we have the gauge invariant strip operators

$$W_{(x)}(x_1, x_2) = \exp\left[ i \int_{x_1}^{x_2} dx \oint dy A_{xy} \right],$$

$$W_{(y)}(y_1, y_2) = \exp\left[ i \oint dx \int_{y_1}^{y_2} dy A_{xy} \right]. \tag{8.9}$$

that generate the $\mathbb{Z}_N$ dipole global symmetry. The exponents in (8.8), (8.9) are quantized because of the periodicity of $\phi^{xy}$ and gauge invariance. These operators satisfy

$$e^{iN\phi^{xy}} = W_{(i)}^N = 1 \tag{8.10}$$

and therefore they are $\mathbb{Z}_N$ operators.

The operators (8.8) and (8.9) do not commute

$$e^{i\phi^{xy}(x,y)} W_{(x)}(x_1, x_2) = e^{2\pi i/N} W_{(x)}(x_1, x_2) e^{i\phi^{xy}(x,y)}, \quad \text{if } x_1 < x < x_2,$$

$$e^{i\phi^{xy}(x,y)} W_{(y)}(y_1, y_2) = e^{2\pi i/N} W_{(y)}(y_1, y_2) e^{i\phi^{xy}(x,y)}, \quad \text{if } y_1 < y < y_2. \tag{8.11}$$

As we will see, the spectrum is in a representation of this Heisenberg-like algebra.[18]

## 8.3 Defects as Fractons

The defects of the $\mathbb{Z}_N$ tensor gauge theory are similar to those in the $U(1)$ tensor gauge theory in Section 7.5. The simplest type of defect is a single static particle

$$\exp\left[ in \int_{-\infty}^{\infty} dt A_0 \right], \quad n = 1, \cdots, N. \tag{8.12}$$

While a single particle cannot move on its own, a pair of them – a dipole – can move collectively along the direction transverse to their separation. The motion of a pair of particles separated in the $x$ direction is described by the defect

$$\exp\left[ in \int_{x_1}^{x_2} dx \int_{\mathcal{C}} \left( dt \partial_x A_0 + dy A_{xy} \right) \right], \quad n = 1, \cdots, N. \tag{8.13}$$

Here $\mathcal{C}$ is a curve in the $(y, t)$ plane. There is a similar defect describing a pair of particles separated in the $y$ direction.

In the special case, where $\mathcal{C}$ is at fixed time it has to be closed. Then this operator is the generator of the symmetry operators (8.9).

---

[18]At the risk of confusing the reader, we would like to point out that this lack of commutativity can be interpreted as a mixed anomaly between these two $\mathbb{Z}_N$ symmetries. See [9] for a related discussion on the relativistic one-form symmetries in the $2+1$-dimensional $\mathbb{Z}_N$ gauge theory.

## 8.4 Lattice Tensor Gauge Theory and the Plaquette Ising Model

The $\mathbb{Z}_N$ tensor gauge theory arises as the continuum limits of two different lattice theories, the $\mathbb{Z}_N$ lattice tensor gauge theory and the $\mathbb{Z}_N$ plaquette Ising model. In this sense, the two lattice models are dual to each other at long distances. This is analogous to the IR duality between the ordinary $1+1$-dimensional $\mathbb{Z}_N$ lattice gauge theory and the $\mathbb{Z}_N$ Ising model.

### *Plaquette Ising Model*

The $\mathbb{Z}_N$ plaquette Ising model (see [45] for a review) is the $\mathbb{Z}_N$ version of the XY-plaquette model in Section 2.1. There is a $\mathbb{Z}_N$ phase $U_s$ and its conjugate momentum $V_s$ at each site. They obey the commutation relation $U_s V_s = e^{2\pi i/N} V_s U_s$. The Hamiltonian includes the plaquette interaction and a transverse field term:

$$H = -K \sum_{\hat{x},\hat{y}} U_{\hat{x},\hat{y}} U_{\hat{x}+1,\hat{y}}^{-1} U_{\hat{x},\hat{y}+1}^{-1} U_{\hat{x}+1,\hat{y}+1} - h \sum_s V_s + c.c.. \tag{8.14}$$

We will assume $h$ to be small.

The conserved charge operators are products of $V_s$ along either the $x$ or $y$ directions:

$$
\begin{aligned}
W_{(x)}(\hat{x}) &= \prod_{\hat{y}=1}^{L^y} V_{\hat{x},\hat{y}}, \\
W_{(y)}(\hat{y}) &= \prod_{\hat{x}=1}^{L^x} V_{\hat{x},\hat{y}}.
\end{aligned}
\tag{8.15}
$$

In the continuum, they become the dipole global symmetry operator (8.9). While the $\mathbb{Z}_N$ dipole symmetry is present on the lattice, the $\mathbb{Z}_N$ electric tensor symmetry (8.8) is broken by the $\sum_s V_s$ term in the Hamiltonian.

### *Lattice Tensor Gauge Theory*

The second lattice model is the $\mathbb{Z}_N$ lattice tensor gauge theory. There is a $\mathbb{Z}_N$ phase variable $U_p$ and its conjugate variable $V_p$ on every plaquette $p$. They obey $U_p V_p = e^{2\pi i/N} V_p U_p$. The gauge transformation $\eta_s$ is a $\mathbb{Z}_N$ phase associated with each site. Under the gauge transformation,

$$U_p \to U_p \, \eta_{\hat{x},\hat{y}} \, \eta_{\hat{x}+1,\hat{y}}^{-1} \, \eta_{\hat{x},\hat{y}+1}^{-1} \, \eta_{\hat{x}+1,\hat{y}+1} \tag{8.16}$$

where the product is over the four sites around the plaquette $p$.

Gauss law sets

$$G_s \equiv \prod_{p \ni s} (V_p)^{\epsilon_p} = 1 \tag{8.17}$$

where the product is an oriented product ($\epsilon_p = \pm 1$) over the four plaquettes $p$ that share a common site $s$. The Hamiltonian is

$$H = -\widetilde{h} \sum_p V_p + c.c., \tag{8.18}$$

with Gauss law imposed by hand.

The conserved charges are the $V_p$ at each plaquette. They become the $\mathbb{Z}_N$ electric tensor symmetry generators (8.8) $e^{i\phi^{xy}}$ in the continuum. While the $\mathbb{Z}_N$ electric tensor symmetry is present on the lattice, the $\mathbb{Z}_N$ dipole symmetry is broken by the Hamiltonian.

Alternatively, we can relax (8.17) and impose Gauss law energetically by adding a term to the Hamiltonian

$$H = -K \sum_s G_s - \widetilde{h} \sum_p V_p + c.c.. \tag{8.19}$$

When $h$ and $\widetilde{h}$ are both zero, we see that (8.19) becomes the Hamiltonian (8.14) of the plaquette Ising model if we dualize the lattice and identify $U_p \leftrightarrow V_s, V_p \leftrightarrow U_s^{-1}$. At long distances, they both flow to the $\mathbb{Z}_N$ tensor gauge theory (8.6).

## 8.5 Ground State Degeneracy

Let us study the ground states of the $\mathbb{Z}_N$ tensor gauge theory from the Lagrangian (8.1). Using the equations of motion (8.3), we can solve all the other fields in terms of $\phi$, and the solution space reduces to

$$\left\{\phi\right\} \,/\, \phi \sim \phi + N\alpha. \tag{8.20}$$

Almost all configurations of $\phi$ can be gauged away completely, except for the winding modes:

$$\phi(t,x,y) = 2\pi \left[ \frac{x}{\ell^x} \sum_\beta W_\beta^y \Theta(y - y_\beta) + \frac{y}{\ell^y} \sum_\alpha W_\alpha^x \Theta(x - x_\alpha) - W \frac{xy}{\ell^x \ell^y} \right]$$
$$W_\alpha^x, W_\beta^y \in \mathbb{Z} \quad , \quad W = \sum_\alpha W_\alpha^x = \sum_\beta W_\beta^y \tag{8.21}$$

If we regularize the space by a lattice, these winding modes are labeled by $L^x + L^y - 1$ integers. Similarly, the gauge parameter $\alpha$ can also have the above winding modes. Therefore, there are $N^{L^x + L^y - 1}$ winding modes that cannot be gauged away with their $W^x$, $W^y$ valued in $\mathbb{Z}_N$. These lead to $N^{L^x + L^y - 1}$ ground states.

Next, we will reproduce the ground state degeneracy using the second presentation (8.6) of the $\mathbb{Z}_N$ tensor gauge theory. In the temporal gauge $A_0 = 0$, the phase space is

$$\left\{ \phi^{xy}(x,y), A_{xy}(x,y) \,\Big|\, \partial_x \partial_y \phi^{xy} = 0, \ A_{xy}(x,y) \sim A_{xy}(x,y) + \partial_x \partial_y \alpha(x,y) \right\}. \tag{8.22}$$

The solution modulo gauge transformations is

$$A_{xy} = \frac{1}{\ell^y} f^x(x) + \frac{1}{\ell^x} f^y(y),$$
$$\phi^{xy} = \hat{f}_x(x) + \hat{f}_y(y) \tag{8.23}$$

The effective Lagrangian for $f$ and $\hat{f}$ is

$$L_{eff} = i \frac{N}{2\pi} \left[ \oint dx \hat{f}_x(t,x) \partial_0 \bar{f}^x(t,x) + \oint dy \hat{f}_y(t,y) \partial_0 \bar{f}^y(t,y) \right], \tag{8.24}$$

where $\bar{f}^x, \bar{f}^y$ are defined in (7.26) subject to the constraint (7.27). The modes $\bar{f}^x, \bar{f}^y$ have delta function periodicities (7.28) and (7.29).

The identification (5.7) implies that the modes $\hat{f}_x, \hat{f}_y$ are pointwise $2\pi$ periodic:

$$\hat{f}_x(x) \sim \hat{f}_x(x) + 2\pi w^x(x),$$
$$\hat{f}_y(y) \sim \hat{f}_y(y) + 2\pi w^y(y), \tag{8.25}$$

where $w^i(x^i) \in \mathbb{Z}$. $\hat{f}_x$ and $\hat{f}_y$ share a common zero mode, which leads to the gauge symmetry

$$
\begin{aligned}
\hat{f}_x(x) &\rightarrow \hat{f}_x(x) + c, \\
\hat{f}_y(y) &\rightarrow \hat{f}_y(y) - c.
\end{aligned}
\tag{8.26}
$$

On a lattice with spacing $a$, we can solve $\bar{f}_y(\hat{y} = L^y)$ in terms of $\bar{f}_x(\hat{x})$ and the other $\bar{f}_y(\hat{y})$ using (7.27). The remaining, unconstrained $L^x + L^y - 1$ $\bar{f}$'s have periodicities $\bar{f}^i(\hat{x}^i) \sim \bar{f}^i(\hat{x}^i) + 2\pi/a$ for each $\hat{x}^i$. On the other hand, we can use the gauge symmetry (8.26) to gauge fix $\hat{f}_y(\hat{y} = L^y) = 0$. The remaining $L^x + L^y - 1$ $\hat{f}$'s have periodicities $\hat{f}_i(\hat{x}^i) \sim \hat{f}_i(\hat{x}^i) + 2\pi$ for each $\hat{x}^i$. The effective Lagrangian is now written in terms of $L^x + L^y - 1$ pairs of $(\hat{f}_i(\hat{x}^i), \bar{f}^i(\hat{x}^i))$:

$$
L_{eff} = i \frac{N}{2\pi} a \left[ \sum_{\hat{x}=1}^{L^x} \hat{f}_x(t, \hat{x}) \partial_0 \bar{f}^x(t, \hat{x}) + \sum_{\hat{y}=1}^{L^y - 1} \hat{f}_y(t, \hat{y}) \partial_0 \bar{f}^y(t, \hat{y}) \right].
\tag{8.27}
$$

Each pair of $(\hat{f}_i(\hat{x}^i), \bar{f}^i(\hat{x}^i))$ leads to an $N$-dimensional Hilbert space.

One way to understand these states is the following. The analysis of the spectrum of the $U(1)$ tensor gauge theory (see Section 7.6) involved $L^x + L^y - 1$ rotors $\bar{f}$, whose quantization led to states carrying $U(1)$ electric tensor symmetry charges. Here, the momentum conjugate to these rotor, $\hat{f}$ is compact and therefore, only charges modulo $N$ are meaningful – states whose charges differ by $N$ are identified. We end up with $N^{L^x + L^y - 1}$ ground states.

The ground state degeneracy can also be understood from the $\mathbb{Z}_N$ global symmetries. On a lattice, the commutation relations between the $\mathbb{Z}_N$ dipole and electric global symmetries (8.11) are isomorphic to $L^x + L^y - 1$ copies of the $\mathbb{Z}_N$ Heisenberg algebra, $AB = e^{2\pi i/N} BA$ and $A^N = B^N = 1$. The isomorphism is given by

$$
\begin{aligned}
A_{\hat{x}} &= e^{i\phi^{xy}(\hat{x},1)}, & B_{\hat{x}} &= W_{(x)}(\hat{x}), & \hat{x} &= 1, \cdots, L^x, \\
A_{\hat{y}} &= e^{i\phi^{xy}(1,\hat{y}) - i\phi^{xy}(1,1)}, & B_{\hat{y}} &= W_{(y)}(\hat{y}), & \hat{y} &= 2, \cdots, L^y,
\end{aligned}
\tag{8.28}
$$

where $W_{(x)}(\hat{x}) \equiv \exp\left[ ia^2 \sum_{\hat{y}=1}^{L^y} A_{xy}(\hat{x}, \hat{y}) \right]$ is a strip operator along the $y$ direction with width $a$, and similarly for $W_{(y)}(\hat{y})$. The minimal representation of the $\mathbb{Z}_N$ Heisenberg algebra is $N$-dimensional. Therefore, the nontrivial algebra (8.11) forces the ground state degeneracy to be $N^{L^x + L^y - 1}$.[19]

## 8.6 Robustness

Let us discuss the robustness of the $\mathbb{Z}_N$ tensor gauge theory. The global symmetry $G_{IR}$ of the low-energy $\mathbb{Z}_N$ tensor gauge theory consists of the $\mathbb{Z}_N$ electric tensor symmetry and the $\mathbb{Z}_N$ winding dipole symmetry

As discussed in Section 8.4, the low-energy $\mathbb{Z}_N$ tensor gauge theory (8.6) can be realized either from the $\mathbb{Z}_N$ Ising plaquette theory, or from the lattice $\mathbb{Z}_N$ tensor gauge theory. In the former short distance realization, the $\mathbb{Z}_N$ dipole symmetry is present on the lattice and will be taken to be our $G_{UV}$. If we impose this microscopic symmetry $G_{UV}$, then there is no $G_{UV}$-invariant relevant operator at long distances that violates $G_{IR}$. Hence $G_{IR}$ is robust.

In fact, there is no $G_{UV}$-invariant local operator at all in the continuum Lagrangian (8.6). The only local operator $e^{i\phi^{xy}}$ is charged under $G_{UV}$, and $\partial_0 \phi^{xy}, \partial_x \partial_y \phi^{xy}$ as well as their derivatives are set to zero by the equations of motion. Therefore, the results obtained from the Lagrangian (8.6) are universal when the global symmetry $G_{UV}$ is imposed.

---

[19]For ordinary $2 + 1$-dimensional $\mathbb{Z}_N$ gauge theory on a 2-torus, the electric and magnetic one-form global symmetries give rise to 2 pairs of $\mathbb{Z}_N$ Heisenberg algebra. Hence the ground state degeneracy is $N^2$.

Instead, if we start with the lattice $\mathbb{Z}_N$ tensor gauge theory, then the $\mathbb{Z}_N$ dipole symmetry is absent at short distances. Since the $\mathbb{Z}_N$ dipole symmetry is not imposed, one is allowed to add local gauge-invariant operators such as $e^{i\phi^{xy}}$ to the Lagrangian. Such perturbations generically lift the ground state degeneracy and break the $\mathbb{Z}_N$ dipole global symmetry explicitly.[20] Hence, the emergent $\mathbb{Z}_N$ dipole symmetry (and therefore $G_{IR}$) is not robust. This is similar to the ordinary $1 + 1$-dimensional $\mathbb{Z}_N$ gauge theory in Section 1.2 (see Table 4 for the analogy).

Finally, our discussion in Section 6 leads to interesting consequences about the phases of the $\mathbb{Z}_N$ Ising plaquette model. Consider the XY-plaquette model close to the continuum limit where we scale $a$ to be parametrically small and the other lattice couplings accordingly, at the same time keeping the system size finite. We perturb the short-distance theory by an operator of the form $e^{iN\phi}$ and thus break the $U(1)$ symmetry to $\mathbb{Z}_N$. When the coefficient of this operator is small enough, we can analyze its effect by perturbing the low-energy theory by the corresponding operator. However, as we discussed in Section 6, this operator is infinitely irrelevant in this range of parameters. As a result, the low-energy theory is not perturbed and it has an emergent $U(1)$ global symmetry. More generally, this means that the $\mathbb{Z}_N$ Ising plaquette model has a range of coupling constants, such that its low-energy behavior is gapless! This gapless theory is described by the continuum theory of Section 2.2. Furthermore, the range of coupling constants with gapless behavior in of co-dimension zero, i.e., it is not fine-tuned. This situation is similar to the existence of a range of parameters with a robust gapless phase in the $1 + 1$-dimensional $\mathbb{Z}_N$ clock models with $N \geq 5$. Note that in our case, this happens for all $N$.

## Acknowledgements

We thank X. Chen, M. Cheng, M. Fisher, A. Gromov, M. Hermele, P.-S. Hsin, A. Kitaev, S. Kivelson, D. Radicevic, L. Radzihovsky, S. Sachdev, D. Simmons-Duffin, S. Shenker, K. Slagle, D. Stanford for helpful discussions. We also thank P. Gorantla, A. Gromov, Z. Komargodski, H.T. Lam, D. Radicevic, T. Rudelius, S. Shenker, and K. Slagle for comments on a draft. The work of N.S. was supported in part by DOE grant DE—SC0009988. NS and SHS were also supported by the Simons Collaboration on Ultra-Quantum Matter, which is a grant from the Simons Foundation (651440, NS). Opinions and conclusions expressed here are those of the authors and do not necessarily reflect the views of funding agencies.

## A  Correlation Functions

In this appendix we consider correlation functions of the continuum theory based on (2.7). In [19], related correlation functions have been analyzed for the microscopic lattice model.

In Euclidean signature, the equation of motion is

$$\mu_0 \partial_\tau^2 \phi - \frac{1}{\mu} \partial_x^2 \partial_y^2 \phi = 0, \tag{A.1}$$

where $\tau$ is the Euclidean time. The two-point function of $\phi$ is

$$\langle \phi(\tau, x, y) \phi(0) \rangle = \frac{1}{(2\pi)^3} \int_{-\infty}^{\infty} d\omega \, dk_x \, dk_y \, \frac{e^{i\omega\tau + ik_x x + ik_y y}}{\mu_0 \omega^2 + \frac{k_x^2 k_y^2}{\mu}}. \tag{A.2}$$

---

[20]In [2], we will discuss the $3 + 1$-dimensional $\mathbb{Z}_N$ tensor gauge theory, which is the low-energy limit of the X-cube model. In this theory, all the gauge-invariant operators are extended objects as opposed to local operators. Therefore the ground state degeneracy and the $\mathbb{Z}_N$ global symmetries are robust.

The $\omega$ integral can be done by deforming the contour and applying the residue theorem:

$$\langle\phi(\tau,x,y)\phi(0)\rangle = \frac{2}{(2\pi)^2}\sqrt{\frac{\mu}{\mu_0}}\int_0^\infty dk_x\int_0^\infty dk_y\frac{e^{-\frac{k_x k_y}{\sqrt{\mu\mu_0}}|\tau|}}{k_x k_y}\cos(k_x x)\cos(k_y y).\tag{A.3}$$

The integrals diverge in the IR region $k_x \to 0$ or $k_y \to 0$ reflecting the many states there. Even if we place the system in finite volume, i.e. on a two-torus of lengths $\ell^x, \ell^y$, the modes with $k_x = 0$ and the modes with $k_y = 0$ lead to a divergence even for generic $\tau, x, y$.

The two point function of $\partial_\tau \phi$ is

$$\langle\partial_\tau\phi(\tau,x,y)\partial_\tau\phi(0)\rangle = -\frac{2}{(2\pi)^2}\frac{1}{\mu^{\frac{1}{2}}\mu_0^{\frac{3}{2}}}\int_0^\infty dk_x\int_0^\infty dk_y\, e^{-\frac{k_x k_y}{\sqrt{\mu\mu_0}}|\tau|}k_x k_y\cos(k_x x)\cos(k_y y).\tag{A.4}$$

It is convergent unless $x = y = 0$.

The case of $x = y = 0$ is interesting because this two point function computes the norm of a state created by acting with $\partial_\tau\phi$ on the vacuum. In order to analyze it more carefully, we place the theory on a two-torus of lengths $\ell^x, \ell^y$. The momenta are quantized, $k_i = 2\pi\frac{n_i}{\ell^i}$ with $n_i \in \mathbb{Z}$. Then, the two point function is a convergent, discrete sum

$$\langle\partial_\tau\phi(\tau,0,0)\partial_\tau\phi(0)\rangle = -\frac{2}{\mu^{\frac{1}{2}}\mu_0^{\frac{3}{2}}}\frac{(2\pi)^2}{(\ell^x\ell^y)^2}\sum_{n_x=0}^\infty\sum_{n_y=0}^\infty n_x n_y e^{-4\pi^2 n_x n_y\frac{|\tau|}{\sqrt{\mu\mu_0}\ell^x\ell^y}}.\tag{A.5}$$

Hence the norm of the state created by $\partial_\tau\phi$ is finite when the space has finite volume. The discrete sum can be evaluated to be (note that only modes with $n_x n_y \neq 0$ contribute)

$$\sum_{n_x,n_y=1}^\infty n_x n_y e^{-n_x n_y T} = \frac{1}{4}\sum_{n=1}^\infty\frac{n}{\sinh(nT/2)^2}\tag{A.6}$$

where $T \equiv \frac{4\pi^2|\tau|}{\sqrt{\mu\mu_0}\ell^x\ell^y}$. When $T$ is small, the sum receives contribution from large $n$, and we can approximate the sum by an integral in $n$, with an IR cutoff at $n = 1$. This gives

$$\sum_{n_x,n_y=1}^\infty n_x n_y e^{-n_x n_y T}\xrightarrow[T\to 0]{}\frac{1}{T^2}\left[\log(2/T)+\mathcal{O}(1)\right].\tag{A.7}$$

Hence in the small $T$ limit (i.e. large torus area compared to $\tau^2$), the norm is[21]

$$\langle\partial_\tau\phi(\tau,0,0)\partial_\tau\phi(0)\rangle = -\frac{2}{(2\pi)^2}\sqrt{\frac{\mu}{\mu_0}}\frac{1}{\tau^2}\left[\log\left(\frac{\sqrt{\mu\mu_0}\ell^x\ell^y}{|\tau|}\right)+\mathcal{O}(1)\right].\tag{A.8}$$

Note that the norm has an IR divergence as we take the area of the torus $\ell^x\ell^y$ to infinity.

Let us study the two point function $\langle e^{i\phi}e^{-i\phi}\rangle$. Since the exponential operators carry the momentum symmetry (2.10), the two operators have to be at the same spatial point

$$\langle e^{i\phi(\tau,0,0)}e^{-i\phi(0)}\rangle = \exp\left(\langle\phi(\tau,0,0)\phi(0)\rangle\right).\tag{A.9}$$

We need to study (A.2) more carefully and regularize it as

$$\langle\phi(\tau,0,0)\phi(0)\rangle = \frac{1}{2\pi\mu_0\ell^x\ell^y}\int_{-\infty}^\infty d\omega\sum_{n_x=0}^{L^x-1}\sum_{n_y=0}^{L^y-1}\frac{e^{i\omega\tau}}{\omega^2+\frac{(2\pi)^4 n_x^2 n_y^2}{\mu\mu_0(\ell^x\ell^y)^2}+\epsilon^2}.\tag{A.10}$$

---

[21]Recall that in Euclidean signature, reflection positivity states that the two-point function of an operator with an index in the time direction $\tau$ is non-positive, i.e., $\langle\mathcal{O}^\tau(-\tau,x^i)\mathcal{O}^\tau(\tau,x^i)\rangle \leq 0$.

Here $\epsilon \to 0$ regularizes the integral over $\omega$ and as above, we placed the system in a box.

The terms with $n_x n_y \neq 0$ are exponentially small at large $|\tau|$ – they are $\mathcal{O}\left(e^{-\frac{(2\pi)^2}{\sqrt{\mu\mu_0\ell^x\ell^y}}|\tau|}\right)$.
So let us focus on the other terms

$$
\begin{aligned}
\langle \phi(\tau,0,0)\phi(0)\rangle &= \frac{1}{2\pi\mu_0\ell^x\ell^y}(L^x+L^y-1)\int_{-\infty}^{\infty} d\omega \frac{e^{i\omega\tau}}{\omega^2+\epsilon^2} + \cdots \\
&= \frac{1}{2\mu_0\ell^x\ell^y}\frac{1}{\epsilon}(L^x+L^y-1) - \frac{1}{2\mu_0\ell^x\ell^y}(L^x+L^y-1)|\tau| + \cdots,
\end{aligned}
\tag{A.11}
$$

where we neglected terms that vanish as $\epsilon \to 0$. Substituting this in (A.9), the first term, which is time independent, can be absorbed in wave function renormalization. The second term leads to exponential decay and is associated with the lowest energy state that contributes to the two point function. Its energy is

$$
\frac{1}{2\mu_0\ell^x\ell^y}\left(\frac{\ell^x}{a} + \frac{\ell^y}{a} - 1\right).
\tag{A.12}
$$

This agrees with the lowest energy state with the quantum numbers of $e^{i\phi}$ (4.14).

This computation also confirms the assertion in Section 6 that the operator $e^{i\phi}$ is highly irrelevant. Repeating this analysis in the dual version of this theory it also confirms the assertion there that $e^{i\phi^{xy}}$ is highly irrelevant.

Next, we consider the properties of the product (6.4), where the two operators are near each other. For simplicity, we set, as in (6.3), $n = -n' = 1$

$$
e^{i\phi(x_0,0)}e^{-i\phi(0,0)}.
\tag{A.13}
$$

As discussed in Section 6, this composite operator is well defined. It might be thought of as a way to define $\partial_x^k \phi$ in the continuum limit, but this interpretation is misleading. The reason is that the ordinary operator product expansion does not apply here. One way to understand it is to note that (A.13) carries a momentum dipole symmetry charge (2.10):

$$
Q^x(x) = \delta(x-x_0) - \delta(x), \qquad Q^y(y) = 0.
\tag{A.14}
$$

We are going to consider the two-point function of (A.13). Using Wick contractions:

$$
\begin{aligned}
&\langle e^{i\phi(\tau,x_0,0)-i\phi(\tau,0,0)}e^{-i\phi(0,x_0,0)+i\phi(0,0,0)}\rangle \\
&\sim \exp\Big[\langle\phi(\tau,x_0,0)\phi(0,x_0,0)\rangle - \langle\phi(\tau,x_0,0)\phi(0,0,0)\rangle \\
&\qquad - \langle\phi(\tau,0,0)\phi(0,x_0,0)\rangle + \langle\phi(\tau,0,0)\phi(0,0,0)\rangle\Big].
\end{aligned}
\tag{A.15}
$$

Here we did not include contractions like $\langle\phi(\tau,x_0,0)\phi(\tau,0,0)\rangle$, which do not affect the $\tau$ dependence of the answer. Hence, the symbol $\sim$ in the equation.

We place the system in finite volume and regularize the UV by a lattice. Then we should consider $x_0 = ma$ with integer $m$. Being interested in the limit of small $x_0$, we take the continuum limit with fixed $m$. The correlation function simplifies in the large $|\tau|$ limit:

$$
\begin{aligned}
&\log\langle e^{i\phi(\tau,x_0,0)-i\phi(\tau,0,0)}e^{-i\phi(0,x_0,0)+i\phi(0,0,0)}\rangle \\
&\sim \frac{1}{2\pi\mu_0\ell^x\ell^y}\left[2L^x - 2 - 2\sum_{n_x=1}^{L^x-1}\cos\left(\frac{2\pi n_x}{\ell^x}x_0\right)\right]\int_{-\infty}^{\infty} d\omega \frac{e^{i\omega\tau}}{\omega^2+\epsilon^2} + \cdots \\
&\sim \frac{1}{2\mu_0\ell^x\ell^y}\frac{2L^x}{\epsilon} - \frac{1}{2\mu_0\ell^x\ell^y}(2L^x)|\tau| + \cdots.
\end{aligned}
\tag{A.16}
$$

The first term is again absorbed into the wavefunction renormalization. The second term is associated with the lowest energy state that contributes to the correlation function. Its energy is

$$\frac{1}{2\mu_0 \ell^x \ell^y} \left( \frac{2\ell^x}{a} \right). \tag{A.17}$$

This agrees with the energy (4.15) for the state with charge (A.14).

The fact that these correlation functions reproduce the energy of these states, is a highly nontrivial check of our treatment of the discontinuous fields and it demonstrates that our analysis is meaningful for these infinite energy states.

These computations also confirm that charged operators of the form $e^{i\phi}$ and $e^{i\phi(\tau,x_0,0)-i\phi(\tau,0,0)}$ are infinitely irrelevant in the continuum limit.

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
