# Peer review of "Exotic Symmetries, Duality, and Fractons in 2+1-Dimensional Quantum Field Theory"

_SciPost Physics, doi:SciPost Phys. 10, 027 (2021)_

## Round 2 · Referee Report · Anonymous · 2020-7-24

Strengths

1- This work provides a path towards large generalizations of the existing fracton phases which is more systematic than
the construction of solvable lattice models:
Pick a subgroup $\Gamma$ of SO(3), put fields in representations of $\Gamma$,
and then follow the rules given in the paper.
(It is not completely systematic in the sense that there are still choices to be made about which symmetries to gauge.)
This is easier to do than constructing an exactly solvable lattice model on a lattice with point group $\Gamma$ (certainly for a high-energy physicist).

2- The discussion of 't Hooft naturaless versus "robustness" (as defined here) is quite useful in clarifying
the extent to which the physics discussed here is generic or robust in a colloquial sense. This understanding is assumed in the condensed matter literature, but is less obvious from the point of view taken in this series of papers.

Weaknesses

1- The business with modes whose energy diverges like $a$ rather than $a^2$ is quite mysterious, as the authors know.
It would be nice to know what is the correct way to think about this.

2- The paper is ungenerous in its treatment of the existing literature. This is a general trend in the paper, but I will give some concrete suggestions for where it can be improved.

-- While the paper of Paramekanti Balents and Fisher is indeed cited at the beginning of the discussion of the model they wrote down, the actual work they did in analyzing that model
(never mind in deducing it as the effective field theory from a lattice model)
is completely ignored, despite the fact that a large part of the work of the present paper recapitulates it in only slightly different language.
Note that here I am not saying that the present authors did not add something important to this discussion (for example, from the 2002 paper it is not at all obvious that there is a deep connection to the fracton phenomena), but perhaps they overestimate the size of the gulf between their language and that of the literature they are harvesting.

-- Within the more recent literature on fractons, there have been several attempts to give a field theory account of the phenomena.
I feel that the work of Pretko, particularly on the role of dipole conservation, has been important for our understanding of the subject, and indeed more important and lasting in its validity than one would infer from reading the present manuscript.

I feel also that the work of Bulmash and Barkeshli did a lot to clarify the essential (if not manifestly universal) ingredients in many fracton constructions from a field theory perspective, including even Haah's cubic code.
(The latter appears in
https://arxiv.org/abs/1806.01855
which is not cited here.)
The innocent reader of this paper would have no idea about the importance of all of this work.

A motivation one could give for minimizing references to previous literature is clarity of exposition.
And a great virtue of this work is its apparently systematic logical development.
However, it would cost the authors nothing to be quite a bit clearer about the important contributions of these papers, for example
in a separate paragraph devoted to this subject.

Report

The motivation for this paper (and the larger program of which it is a part)
comes from the challenge to the inevitability of continuum field theory as a
description of the low energy physics of local lattice models
associated with fracton topological phases.
I agree fully that this is an important issue to clarify.

It is not entirely clear yet whether the kinds of generalization of the rules of QFT the authors contemplate here are something which we would want to think about for other reasons besides reproducing the low energy data of these lattice models, as the authors acknowledge. Nevertheless I think it is a useful step.

The paper should be published.

Requested changes

1- Please improve the discussion of previous literature, at least along the lines described above.

2- page 6
"Then, we deform slightly the short distance parameters and ask whether the low-energy system still preserve"
should be "... still preserves"

---

## Editorial Decision

published